# REGULARIZATION MATTERS IN POLICY OPTIMIZATION

## ABSTRACT

Deep Reinforcement Learning (Deep RL) has been receiving increasingly more attention thanks to its encouraging performance on a variety of control tasks. Yet, conventional regularization techniques in training neural networks (e.g., $L_2$ regularization, dropout) have been largely ignored in RL methods, possibly because agents are typically trained and evaluated in the same environment. In this work, we present the first comprehensive study of regularization techniques with multiple policy optimization algorithms on continuous control tasks. Interestingly, we find conventional regularization techniques on the policy networks can often bring large improvements on the task performance, and the improvement is typically more significant when the task is more difficult. We also compare with the widely used entropy regularization and find $L_2$ regularization is generally better. Our findings are further confirmed to be robust against the choice of training hyperparameters. We also study the effects of regularizing different components and find that only regularizing the policy network is typically the best option. We hope our study provides guidance for future practices in regularizing policy optimization algorithms.

## 1 INTRODUCTION

Regularization, typically referring to methods for preventing overfitting, is a key technique in successfully training a neural network. Perhaps the most widely recognized regularization methods in deep learning are $L_2$ regularization (also known as weight decay) and dropout (Srivastava et al., 2014). Those techniques are standard practices in supervised learning tasks from many domains. Major tasks in computer vision, e.g., image classification (He et al., 2016; Huang et al., 2017), object detection (Ren et al., 2015; Redmon et al., 2016), all use $L_2$ regularization as a default option. In natural language processing, for example, the Transformer model (Vaswani et al., 2017) uses dropout. and the recently popular BERT model (Devlin et al., 2018) uses $L_2$ regularization. In fact, it is very rare to see state-of-the-art neural models trained without any regularization in a supervised setting.

However, in deep reinforcement learning (RL), those conventional regularization methods are largely absent or underutilized in past research, possibly because in most cases we are maximizing the return on exactly the same task as in training. In other words, there is a lack of generalization gap from the training environment to the test environment (Cobbe et al., 2018). Moreover, researchers in deep RL focus more on high-level algorithm designs, which is more closely related to the field of reinforcement learning, and focus less on network training techniques such as regularization. For popular policy optimization algorithms like Asynchronous Advantage Actor-Crtic (A3C) (Mnih et al., 2016), Trust Region Policy Optimization (TRPO) (Schulman et al., 2015), Proximal Policy Optimization (PPO) (Schulman et al., 2017), and Soft Actor Critic (SAC) (Haarnoja et al., 2018), conventional regularization methods were not considered. Even in popular codebases such as the OpenAI Baseline (Dhariwal et al., 2017), $L_2$ regularization and dropout were not incorporated.

Instead, the most commonly used regularization in the RL community, is an "entropy regularization" term that penalizes the high-certainty output from the policy network, to encourage more exploration during the training process and prevent the agent from overfitting to certain actions. The entropy regularization was first introduced by Williams & Peng (1991) and now used by many contemporary algorithms (Mnih et al., 2016; Schulman et al., 2017; Teh et al., 2017; Farebrother et al., 2018).

In this work, we take an empirical approach to questioning the conventional wisdom of not using common regularizations. We study agent's performance on the current task (the environment which the agent is trained on), rather than its generalization ability to different environments as many recent works (Zhang et al., 2018a; Zhao et al., 2019; Farebrother et al., 2018; Cobbe et al., 2018). We specifically focus our study on policy optimization methods, which are becoming increasingly popular and have achieved top performance on various tasks. We evaluate four popular policy optimization algorithms, namely SAC, PPO, TRPO, and the synchronous version of Advantage Actor Critic (A2C), on multiple continuous control tasks. A variety of conventional regularization techniques are considered, including $L_2/L_1$ weight regularization, dropout, weight clipping (Arjovsky et al., 2017) and Batch Normalization (BN) (Ioffe & Szegedy, 2015). We compare the performance of these regularization techniques to that without regularization, as well as the entropy regularization.

Surprisingly, even though the training and testing environments are the same, we find that many of the conventional regularization techniques, when imposed to the policy networks, can still bring up the performance, sometimes significantly. Among those regularizers, $L_2$ regularization, perhaps the most simple one, tends to be the most effective for all algorithms and generally outperforms entropy regularization. $L_1$ regularization and weight clipping can boost performance in many cases. Dropout and Batch Normalization tend to bring improvements only on off-policy algorithms. Additionally, all regularization methods tend to be more effective on more difficult tasks. We also verify our findings with a wide range of training hyperparameters and network sizes, and the result suggests find that imposing proper regularization can sometimes save the effort of tuning other training hyperparameters. Finally, we study which part of the policy optimization system should be regularized, and conclude that generally only regularizing the policy network suffices, as imposing regularization on value networks usually does not help. Our results also show that neural network training techniques such as regularization, can be as important as high-level reinforcement learning algorithms in terms of boosting performance. Our main contributions can be summarized as follows:

- We provide the first comprehensive study of common regularization methods in policy optimization algorithms, which have been largely ignored in the RL literature.

- We find conventional regularizations can often be very effective in improving the performance on continuous control tasks, espcially on harder ones. Remarkably, the most simple $L_2$ regularization generally performs better than the more widely used entropy regularization. BN and dropout can only help in off-policy algorithms.

- We experiment with multiple randomly sampled training hyperparameters for each algorithm and confirm our findings still hold. The result also suggests that proper regularization can sometimes ease the hyperparameter tuning process.

- We study which part of the network(s) should be regularized. The key lesson is to regularize the policy network but not the value network.

## 2  RELATED WORKS

**Regularization in Deep RL.** Conventional regularization methods have rarely been applied in deep RL. One rare case of such use is in Deep Deterministic Policy Gradient (DDPG) (Lillicrap et al., 2016), where Batch Normalization is applied to all layers of the actor $\mu$ network and layers of the critic $Q$ network prior to the action input, and $L_2$ regularization is applied to the critic $Q$ network because it tends to have overestimation bias (Fujimoto et al., 2018).

Some recent studies have developed more complicated regularization approaches to continuous control tasks. Cheng et al. (2019) regularizes the stochastic action distribution $\pi(a|s)$ using a suboptimal control prior. The regularization weight at a given state is adjusted based on the temporal difference (TD) error. The larger the TD error, the more the action distribution moves towards the prior. Galashov et al. (2019) introduces a default policy that receives limited information as a regularizer. The information asymmetry between the behavior policy and the default policy helps to accelerate convergence and improve performance. Parisi et al. (2019) introduces TD error regularization to penalize inaccurate value estimation and Generalized Advantage Estimation (GAE) (Schulman et al., 2016) regularization to penalize GAE variance. However, most of these regularizations are rather complicated (Galashov et al., 2019), specifically designed for certain algorithms (Parisi et al., 2019), or need prior information (Cheng et al., 2019). Also, these techniques consider regularizing the output

of the network, while conventional regularization methods mostly directly regularize the parameters. In this work, we focus on studying these simpler but under-utilized regularization methods.

**Generalization in Deep RL** typically refers to how the model perform in a different environment from the one it is trained on. The generalization gap can come from different modes/levels/difficulties of a game (Farebrother et al., 2018), simulation vs. real world (Tobin et al., 2017), parameter variations (Pattanaik et al., 2018), or different random seeds in environment generation (Zhang et al., 2018b). There are a number of methods designed to address this issue, e.g., through training the agent over multiple domains/tasks (Tobin et al., 2017; Rajeswaran et al., 2017), adversarial training (Tobin et al., 2017), designing model architectures (Srouji et al., 2018), adaptive training (Duan et al., 2016), etc. Meta Reinforcement Learning (Finn et al., 2017; Gupta et al., 2018; Al-Shedivat et al., 2017) try to learn generalizable agents by training on a set of environments drawn from the same family/distribution. There are also some comprehensive studies on RL generalization with interesting findings (Zhang et al., 2018a;b; Zhao et al., 2019; Packer et al., 2018), e.g., algorithms performing better on the training environment could perform worse under domain shift (Zhao et al., 2019).

Recently, several studies have investigated conventional regularization's effect on generalization. Farebrother et al. (2018) shows that in Deep Q-Networks (DQN), $L_2$ regularization and dropout can sometimes bring benefit when evaluated on the same Atari game with mode and difficulty variations. Cobbe et al. (2018) shows that $L_2$ regularization, dropout, data augmentation, and Batch Normalization can improve generalization performance, but to a less extent than entropy regularization and $\epsilon$-greedy action selection, when evaluated with (Espeholt et al., 2018). Different from those studies, we focus on regularization's effect in the same environment, a more direct goal compared with generalization, yet on which conventional regularizations are under-explored.

## 3 REGULARIZATION METHODS

There are in general two kinds of common approaches for imposing regularization, one is to discouraging complex models (e.g., weight regularization, weight clipping), and the other is to inject certain kind of noise in the activations (e.g., dropout and Batch Normalization). Here we briefly introduce those regularization methods we investigate in our experiments.

$L_2$ / $L_1$ **Weight Regularization.** Large weights are usually believed to be a sign of overfitting to the training data, since the function it represents tend to be complex. One can encourage small weights by adding a loss term penalizing the norm of the weight vector. Suppose $L$ is the original empirical loss we want to minimize. SGD updates the model on a mini-batch of training samples: $\theta_i \leftarrow \theta_i - \eta \cdot \frac{\partial L}{\partial \theta_i}$, where $\eta$ is the learning rate. When applying $L_2$ regularization, we add an additional $L_2$-norm squared loss term $\frac{1}{2}\lambda||\theta||_2^2$ to the training objective. Thus the SGD step becomes $\theta_i \leftarrow \theta_i - \eta \frac{\partial L}{\partial \theta_i} - \eta \cdot \lambda \cdot \theta_i$. Similarly, in the case of $L_1$ weight regularization, the additional loss term is $\lambda||\theta||_1$, and the SGD step becomes $\theta_i \leftarrow \theta_i - \eta \cdot \frac{\partial L}{\partial \theta_i} - \eta \cdot \lambda \cdot \text{sign}(\theta_i)$.

**Weight Clipping.** Weight clipping is an extremely simple idea: after each gradient update step, each individual weight is clipped to range $[-c, c]$, where $c$ is a hyperparameter. This could be formally described as $\theta_i \leftarrow \max(\min(\theta_i, c), -c)$. In Wasserstein GANs (Arjovsky et al., 2017), weight clipping is used to satisfy the constraint of Lipschitz continuity. This plays an important role in stabilizing the training of GANs (Goodfellow et al., 2014), which were notoriously hard to train and often suffered from "mode collapse" before. Weight clipping could also be seen as a regularizor since it drastically reduce the complexity of the model space, by preventing any weight's magnitude from being larger than $c$.

**Dropout.** Dropout (Srivastava et al., 2014) is one of the most successful regularization techniques developed specifically for neural networks. The idea is to randomly deactivate a certain percentage of neurons during training; during testing, a rescaling operation is taken to ensure the scale of the activations is the same as training. One explanation for its effectiveness in reducing overfitting is they can prevent "co-adaptation" of neurons. Another explanation is that dropout acts as a implicit model ensemble method, because during training a different model is sampled to fit each mini-batch of data.

**Batch Normalization.** Batch Normalization (BN) (Ioffe & Szegedy, 2015) is invented to address the problem of "internal covariate shift", and it does the following transformation: $\hat{z} = \frac{z_{in} - \mu_\mathcal{B}}{\sqrt{\sigma_\mathcal{B}^2 + \epsilon}}$; $z_{out} = \gamma\hat{z} + \beta$, where $\mu_\mathcal{B}$ and $\sigma_\mathcal{B}$ are the mean and standard deviation values of input activations over $\mathcal{B}$, $\gamma$

and $\beta$ are trainable affine transformation parameters (scale and shift) which provides the possibility of linearly transforming normalized activations back to any scales. BN turns out to be able to greatly accelerate the convergence and bring up the accuracy. It has become a standard component, especially in convolutional networks. BN also "acts as a regularizer" (Ioffe & Szegedy, 2015): since the statistics $\mu_{\mathcal{B}}$ and $\sigma_{\mathcal{B}}$ are dependent on the current batch, BN subtracts and divides different values in each iteration. This randomness can encourage subsequent layers to be robust to such variation of input.

**Entropy Regularization.** In a policy optimization framework, the policy network is used to model a conditional distribution over actions, and entropy regularization is widely used to prevent the learned policy from overfitting to one or some of the actions. More specifically, in each step, the output distribution of the policy network is penalized to have a high entropy. Policy entropy is calculated at each step as $H_{s_i} = -\mathbb{E}_{a_i \sim \pi(a_i|s_i)} \log \pi(a_i|s_i)$, where $(s_i, a_i)$ is the state-action pair. Then the per-sample entropy is averaged within the batch of state-action pairs to get the regularization term $L^H = \frac{1}{N} \sum_{s_i} H_{s_i}$. A coefficient $\lambda$ is also needed, and $\lambda L^H$ is added to the policy objective $J(\theta)$ to be maximized during policy updates. Entropy regularization also encourages exploration due to increased stochasticity in actions, leading to better performance in the long run.

## 4 EXPERIMENTS

### 4.1 SETTINGS

**Algorithms.** We evaluate the six regularization methods introduced in Section 3 using four popular policy optimization algorithms, namely, A2C (Mnih et al., 2016), TRPO (Schulman et al., 2015), PPO (Schulman et al., 2017), and SAC (Haarnoja et al., 2018). The first three algorithms are on-policy while the last one is off-policy. For the first three algorithms, we adopt the code from OpenAI Baseline (Dhariwal et al., 2017), and for SAC, we use the official implementation at (Haarnoja, 2018).

**Tasks.** The algorithms with different regularizations are tested on nine continuous control tasks: Hopper, Walker, HalfCheetah, Ant, Humanoid, and HumanoidStandup from the MuJoCo simulation environment (Todorov et al., 2012); Humanoid, AtlasForwardWalk, and HumanoidFlagrun from the more challenging RoboSchool (OpenAI) suite. Among the MuJoCo tasks, agents for Hopper, Walker, and HalfCheetah are easier to learn, while Ant, Humanoid, HumanoidStandup are relatively harder (larger state-action space, more training examples). The three Roboschool tasks are even harder than all the MuJoCo tasks as they require more timesteps to converge. To better understand how different regularization methods work on different difficulties, we roughly categorize the first three environments as "easy" tasks and the last six as "hard" tasks.

**Training.** On MuJoCo tasks, we keep all training hyperparameters unchanged as in the codebase adopted. Since hyperparameters for the RoboSchool tasks are not included in the original codebase, we briefly tune the hyperparameters for each algorithm before we apply any regularization (more details in Appendix D). For details on regularization strength tuning, please refer to Appendix B.

The results shown in this section are obtained by **only regularizing the policy network**, and a further study on this issue will be presented in Section 6. We run each experiment independently with five random seeds, then use the average return over the last 100 episodes as the final result. Each regularization method is evaluated independently, with other regularizations turned off. We refer to the result without any regularization methods as the baseline. For BN and dropout, we use its training mode when we update the network, and test mode when sampling trajectories.

Note that entropy regularization is still applicable for SAC, despite it already incorporates the maximization of entropy in the reward term. In our experiments, we add the entropy regularization term to the policy optimization loss function in equation (12) of the original paper (Haarnoja et al., 2018). Meanwhile, policy network dropout is not applicable to TRPO because during policy updates, different neurons in the old and new policy networks are dropped out, causing different shifts in the old and new action distributions given the same state, which then causes the trust region constraint to be violated. In this case, the algorithm fails to perform any policy update from network initialization.

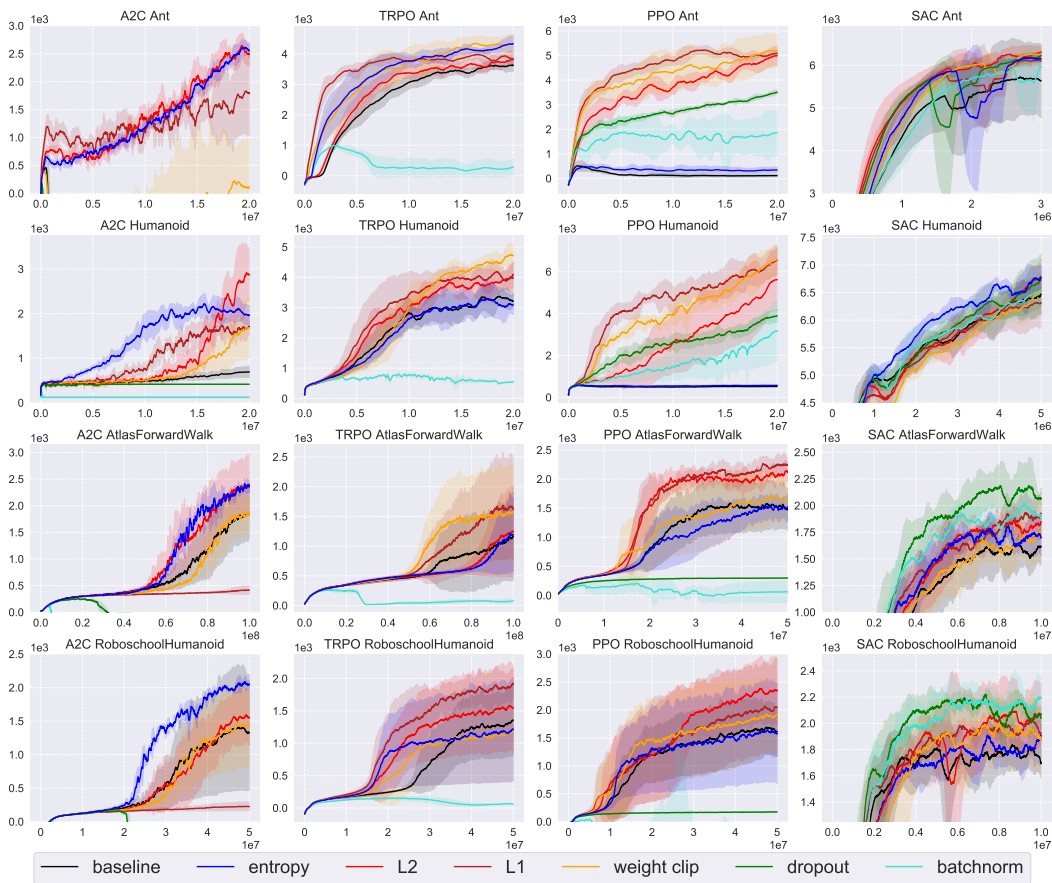

Figure 1: Reward vs. timesteps, for four algorithms (columns) and four environments (rows).

## 4.2 RESULTS.

**Training curves.** We plot the training curves from four environments (rows) in Figure 1, on four algorithms (columns). Figures for the rest five environments are deferred to Appendix C. In the figure, different colors are used to denote different regularization methods, e.g., black is the baseline method. Shades are used to denote ±1 standard deviation range. Notably, these conventional regularizations can frequently boost the performance across different tasks and algorithms, demonstrating that a study on the regularization in deep RL is highly demanding. Interestingly, in some cases where the baseline (with the default hyperparameters in the codebase) does not converge to a reasonable solution, e.g., A2C Ant, PPO Humanoid, imposing some regularization can make the training converge to a high level. Another observation is that BN always significantly hurts the baseline for on-policy algorithms. The reason will be discussed later. For the off-policy SAC algorithm, dropout and BN sometimes bring large improvements on hard tasks like AtlasForwardWalk and RoboschoolHumanoid.

**How often do regularizations help?** To quantitatively measure the effectiveness of the regularizations on each algorithm across different tasks, we **define the condition when a regularization is said to "improve" upon the baseline** in a certain environment. Denote the baseline mean return over five seeds on an environment as $\mu_{\text{env},b}$, and the mean and standard deviation of the return obtained with a certain regularization method over five seeds as $\mu_{\text{env},r}$ and $\sigma_{\text{env},r}$. We say the performance is "improved" by the regularization if $\mu_{\text{env},r} - \sigma_{\text{env},r} > \max(\mu_{\text{env},b}, T(\text{env}))$, where $T(\text{env})$ is the minimum return threshold of an environment. The threshold serves to ensure the return is at least in a reasonable level. We set the threshold to be $10^5$ for HumanoidStandup and $10^3$ for all other tasks.

The result is shown in Table 1. Perhaps the most significant observation is that $L_2$ regularization is the most often to improve upon the baseline. A2C algorithm is an exception, where entropy regularization is the most effective. $L_1$ regularization behaves similar to $L_2$ regularization, but is outperformed

by the latter. Weight clipping's usefulness is highly dependent on the algorithms and environments. Despite in total it only helps at 30.6% times, it can sometimes outperform entropy regularization by a large margin, e.g., in TRPO Humanoid and PPO Humanoid as shown in Figure 1. BN is not useful at all in the three on-policy algorithms (A2C, TRPO, and PPO). Dropout is not useful in A2C at all, and sometimes helps in PPO. However, BN and dropout can be useful in SAC. All regularization methods generally improve more often when they are used on harder tasks, perhaps because for easier ones the baseline is often sufficiently strong to reach a high performance.

It should be noted that under our definition, not "improving" does not indicate the regularization is hurting the performance. If we define "hurting" as $\mu_{\text{env},r} + \sigma_{\text{env},r} < \mu_{\text{env},b}$ (the reward minimum threshold is not considered here), then total percentage of hurting is 0.0% for $L_2$, 2.8% for $L_1$, 5.6% for weight clipping, 44.4% for dropout, 66.7% for BN, and 0.0% for entropy. In other words, under our parameter tuning range, $L_2$ and entropy regularization never hurt with appropriate strengths. For BN and dropout, we also note that almost all hurting cases are in on-policy algorithms, except one case for BN in SAC. If we define "hurting" as $\mu_{\text{env},r} < \mu_{\text{env},b}$, the total percentage of hurting is 11.1% for $L_2$, 16.7% for $L_1$, 22.2% for weight clipping, 55.5% for dropout, 72.2% for BN, and 16.7% for entropy. In sum, all regularizations in our study very rarely hurt the performance except for BN/dropout in on-policy methods.

| Reg \ Alg | A2C | | | TRPO | | | PPO | | | SAC | | | TOTAL | | |
|---|---|---|---|---|---|---|---|---|---|---|---|---|---|---|---|
| | Easy | Hard | Total | Easy | Hard | Total | Easy | Hard | Total | Easy | Hard | Total | Easy | Hard | Total |
| Entropy | **33.3** | **100.0** | **77.8** | 0.0 | 50.0 | 33.3 | 0.0 | 33.3 | 22.2 | 33.3 | 50.0 | 44.4 | 16.7 | 58.3 | 44.4 |
| $L_2$ | 0.0 | 50.0 | 33.3 | 0.0 | **66.7** | **44.4** | **33.3** | **83.3** | **66.7** | **66.7** | **66.7** | **66.7** | 25.0 | **66.7** | **52.8** |
| $L_1$ | 0.0 | 50.0 | 33.3 | 0.0 | **66.7** | **44.4** | **33.3** | 66.7 | 55.6 | 33.3 | 50.0 | 44.4 | 16.7 | 58.3 | 44.4 |
| Weight Clip | 0.0 | 16.7 | 11.1 | **33.3** | 33.3 | 33.3 | **33.3** | 66.7 | 55.6 | 33.3 | 16.7 | 22.2 | 25.0 | 33.3 | 30.6 |
| Dropout | 0.0 | 0.0 | 0.0 | N/A | N/A | N/A | **33.3** | 50.0 | 44.4 | **66.7** | 50.0 | 55.6 | **33.3** | 33.3 | 33.3 |
| BatchNorm | 0.0 | 0.0 | 0.0 | 0.0 | 0.0 | 0.0 | 0.0 | 16.7 | 11.1 | 33.3 | 50.0 | 44.4 | 8.3 | 16.7 | 13.9 |

Table 1: Percentage (%) of environments where the final performance "improves" when using regularization, according to our definition in Section 4.2.

**Ranking all regularizations.** Furthermore, to better compare their relative effectiveness, we rank the performance of all the regularization methods, together with the baseline, for each algorithm and task, and present the average ranks and the standard deviation of ranks in Table 2 and Table 3. Here, the ranks of returns among different regularizers are collected for each environment (after averaging over 5 random seeds), and then the mean and standard deviations are calculated. From Table 2, we observe that, except for BN and dropout in on-policy algorithms, all regularizations on average outperform baselines. Again, $L_2$ regularization is the strongest in most cases. Other similar observations can be made as in Table 1. For every algorithm, baseline ranks lower on harder tasks than easier ones; in total, it ranks 3.50 for easier tasks and 5.25 for harder tasks. This indicates that regularization is more effective when the tasks are harder.

| Reg \ Alg | A2C | | | TRPO | | | PPO | | | SAC | | | TOTAL | | |
|---|---|---|---|---|---|---|---|---|---|---|---|---|---|---|---|
| | Easy | Hard | Total | Easy | Hard | Total | Easy | Hard | Total | Easy | Hard | Total | Easy | Hard | Total |
| Baseline | 3.33 | 4.50 | 4.11 | 3.33 | 4.67 | 4.22 | 3.00 | 6.00 | 5.00 | 4.33 | 5.83 | 5.33 | 3.50 | 5.25 | 4.67 |
| Entropy | **1.00** | **1.50** | **1.33** | 4.67 | 3.00 | 3.56 | **3.00** | 4.17 | 3.78 | **3.00** | 3.83 | 3.55 | 2.92 | 3.13 | 3.06 |
| $L_2$ | 2.67 | **1.50** | 1.89 | **1.33** | 2.83 | **2.33** | **3.00** | 2.17 | **2.45** | **3.00** | 2.67 | **2.78** | **2.50** | **2.29** | **2.36** |
| $L_1$ | 4.33 | 3.67 | 3.89 | 2.67 | **2.17** | 2.34 | 3.33 | 2.67 | 2.89 | 3.67 | 4.83 | 4.44 | 3.50 | 3.34 | 3.39 |
| Weight Clip | 3.67 | 3.83 | 3.78 | 3.00 | 2.33 | 2.55 | **3.00** | 2.50 | 2.67 | 4.33 | 4.17 | 4.22 | 3.50 | 3.21 | 3.31 |
| Dropout | 6.00 | 6.00 | 6.00 | N/A | N/A | N/A | 5.67 | 4.67 | 5.00 | 3.33 | 3.17 | 3.22 | 5.00 | 4.61 | 4.74 |
| BatchNorm | 7.00 | 7.00 | 7.00 | 6.00 | 6.00 | 6.00 | 7.00 | 5.83 | 6.22 | 6.33 | 3.50 | 4.44 | 6.58 | 5.58 | 5.92 |

Table 2: The average rank in the mean return for different regularization methods. $L_2$ regularization tops the ranking for most algorithms and environment difficulties.

| Reg \ Alg | A2C | | | TRPO | | | PPO | | | SAC | | | TOTAL | | |
|---|---|---|---|---|---|---|---|---|---|---|---|---|---|---|---|
| | Easy | Hard | Total | Easy | Hard | Total | Easy | Hard | Total | Easy | Hard | Total | Easy | Hard | Total |
| Baseline | 1.25 | 0.76 | 1.10 | 1.25 | 0.75 | 1.13 | 2.16 | 1.00 | 2.05 | 2.49 | 1.77 | 2.16 | 1.94 | 1.33 | 1.76 |
| Entropy | 0.00 | 0.50 | 0.47 | 0.47 | 1.15 | 1.26 | 0.82 | 2.03 | 1.81 | 2.16 | 2.11 | 2.17 | 1.75 | 1.90 | 1.86 |
| $L_2$ | 0.94 | 0.50 | 0.87 | 0.47 | 1.07 | 1.15 | 1.63 | 0.69 | 1.17 | 0.82 | 1.70 | 1.47 | 1.26 | 1.21 | 1.23 |
| $L_1$ | 0.94 | 0.75 | 0.87 | 0.47 | 1.07 | 0.94 | 1.70 | 0.94 | 1.29 | 0.94 | 1.21 | 1.26 | 1.26 | 1.43 | 1.38 |
| Weight Clip | 0.47 | 0.69 | 0.63 | 1.41 | 1.37 | 1.42 | 0.82 | 1.26 | 1.15 | 2.05 | 1.34 | 1.62 | 1.44 | 1.44 | 1.45 |
| Dropout | 0.00 | 0.00 | 0.00 | N/A | N/A | N/A | 0.47 | 1.97 | 1.70 | 1.70 | 2.11 | 1.99 | 1.61 | 2.04 | 1.91 |
| BatchNorm | 0.00 | 0.00 | 0.00 | 0.00 | 0.00 | 0.00 | 0.00 | 0.90 | 0.92 | 0.47 | 1.71 | 1.95 | 0.49 | 1.61 | 1.42 |

Table 3: The standard deviation of the ranks for different regularization methods. $L_2$, $L_1$ and weight clipping mostly have slightly smaller standard deviations.

## 5 ROBUSTNESS WITH HYPERPARAMETER CHANGES

In the previous section, the experiments are conducted mostly with the default hyperparameters in the codebase we adopt, which are not necessarily optimized. For example, PPO Humanoid baseline performs poorly using default hyperparameters, not converging to a reasonable solution. Meanwhile, it is known that RL algorithms are very sensitive to hyperparameter changes (Henderson et al., 2018). Thus, our findings can be vulnerable to such variations. To further confirm our findings, we evaluate the regularizations under a variety of hyperparameter settings. For each algorithm, we sample five hyperparameter settings for the baseline and apply regularization on each of them. Due to the heavy computation budget, we only evaluate on five MuJoCo environments: Hopper, Walker, Ant, Humanoid, and HumanoidStandup. Under our sampled hyperparameters, poor baselines are mostly significantly improved. For further details on sampling and training curves, please refer to Appendix E and K.

| Reg \ Alg | A2C | | | TRPO | | | PPO | | | SAC | | | TOTAL | | |
|---|---|---|---|---|---|---|---|---|---|---|---|---|---|---|---|
| | Easy | Hard | Total | Easy | Hard | Total | Easy | Hard | Total | Easy | Hard | Total | Easy | Hard | Total |
| Baseline | **2.70** | 4.13 | 3.65 | 3.70 | 3.40 | 3.50 | 3.00 | 5.53 | 4.69 | 4.20 | 5.00 | 4.73 | 3.40 | 4.52 | 4.14 |
| Entropy | 3.50 | 2.93 | 3.12 | 3.60 | 3.47 | 3.51 | 4.30 | 4.40 | 4.37 | **3.10** | 4.47 | 4.01 | 3.63 | 3.82 | 3.75 |
| $L_2$ | 4.40 | **2.27** | 2.98 | 2.50 | 2.53 | **2.52** | **1.90** | **1.80** | **1.83** | 3.50 | **2.73** | **2.99** | **3.08** | **2.33** | **2.58** |
| $L_1$ | **2.70** | 2.53 | **2.59** | 3.10 | **2.27** | 2.55 | 2.80 | 2.20 | 2.40 | 3.70 | 4.00 | 3.90 | **3.08** | 2.75 | 2.86 |
| Weight Clip | 3.30 | 3.13 | 3.19 | **2.20** | 3.33 | 2.95 | 3.70 | 2.87 | 3.15 | 5.80 | 4.27 | 4.78 | 3.75 | 3.40 | 3.52 |
| Dropout | 4.40 | 6.07 | 5.51 | N/A | N/A | N/A | 6.10 | 5.33 | 5.59 | 4.20 | 4.27 | 4.25 | 4.90 | 5.22 | 5.12 |
| BatchNorm | 7.00 | 6.93 | 6.95 | 5.90 | 6.00 | 5.97 | 6.20 | 5.80 | 5.93 | 3.50 | 3.27 | 3.35 | 5.65 | 5.50 | 5.55 |

Table 4: The average rank in the mean return for different regularization methods, under five randomly sampled training hyperparameters for each algorithm.

| Reg \ Alg | A2C | | | TRPO | | | PPO | | | SAC | | | TOTAL | | |
|---|---|---|---|---|---|---|---|---|---|---|---|---|---|---|---|
| | Easy | Hard | Total | Easy | Hard | Total | Easy | Hard | Total | Easy | Hard | Total | Easy | Hard | Total |
| Baseline | 1.17 | 1.29 | 1.39 | 1.40 | 1.44 | 1.42 | 1.26 | 1.45 | 1.81 | 1.60 | 1.93 | 2.16 | 1.48 | 1.69 | 1.85 |
| Entropy | 1.20 | 1.46 | 1.38 | 1.55 | 1.30 | 1.42 | 1.40 | 1.70 | 1.59 | 2.43 | 2.03 | 2.17 | 1.76 | 1.76 | 2.30 |
| $L_2$ | 1.62 | 1.12 | 1.70 | 1.08 | 0.87 | 0.96 | 1.47 | 0.85 | 1.16 | 1.28 | 1.95 | 1.47 | 1.66 | 1.33 | 1.75 |
| $L_1$ | 1.45 | 1.31 | 1.38 | 0.80 | 0.93 | 1.04 | 1.56 | 0.98 | 1.25 | 2.00 | 1.67 | 1.26 | 1.59 | 1.46 | 1.82 |
| Weight Clip | 1.51 | 1.15 | 1.31 | 1.37 | 1.70 | 1.69 | 1.08 | 0.89 | 1.05 | 1.78 | 1.57 | 1.62 | 1.99 | 1.45 | 1.82 |
| Dropout | 2.24 | 0.25 | 1.65 | N/A | N/A | N/A | 1.25 | 1.40 | 1.40 | 1.54 | 1.77 | 1.99 | 1.91 | 1.51 | 1.68 |
| BatchNorm | 0.00 | 0.25 | 0.20 | 0.60 | 0.00 | 0.39 | 0.75 | 1.15 | 1.02 | 1.91 | 2.11 | 1.95 | 1.68 | 1.82 | 2.04 |

Table 5: The standard deviation of the ranks for different regularization methods, under five randomly sampled training hyperparameters.

Similar to Table 2 and Table 3, the results of regularization ranks are shown in Table 4 and Table 5. For results of improvement percentages similar to Table 1, please refer to Appendix F. We note that our main findings still hold: 1) the regularizations can improve more effectively on baselines with harder tasks; 2) $L_2$ is still generally the best regularization method; 3) BN and dropout hurts on-policy algorithms but can bring improvement only for the off-policy SAC algorithm. Interestingly, different from previous section, $L_1$ regularization and weight clipping tend to be more effective than

the entropy regularization. The gaps between entropy and $L_2$, $L_1$ and weight clipping are even larger for harder tasks.

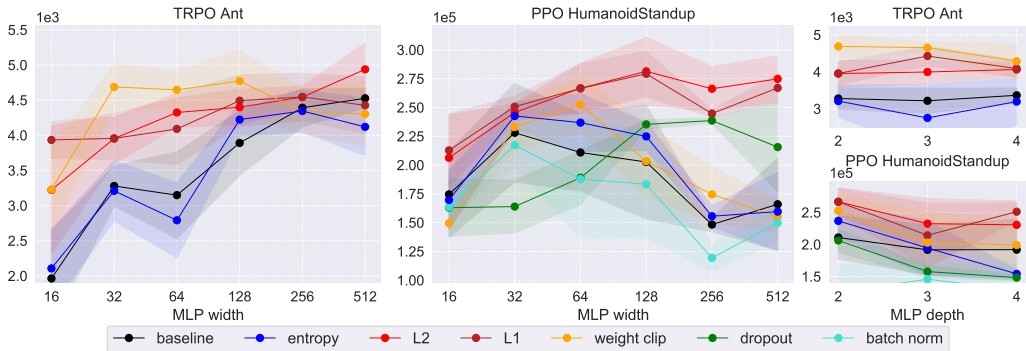

Figure 2: Final reward vs. single hyperparameter change. "Rollout Timesteps" refers to the number of state-action samples used for training between policy updates.

To better visualize the robustness against change of hyperparameters, we show the result when a single hyperparameter is varied in Figure 2. We note that the certain regularizations can consistently improve the baseline with different hyperparameters. In these cases, proper regularizations can ease the hyperparameter tuning process, as they can bring up the performance of baselines with suboptimal hyperparameters to be even higher than baselines with better hyperparameters.

Figure 3: Final reward vs. changes in the width and depth of network.

We also analyze regularizations' effect with different network width/depths in Figure 3. There are several observations we can draw: 1) The baseline performance can be either increasing, decreasing or staying roughly the same when the network increases depth/width. 2) Certain regularizations can help with various widths or depths, demonstrating their robustness against these hyperparameters and ability to ease hyperparameter tuning. 3) Regularizations do not necessarily bring larger improvement when the network sizes are bigger, contrary to what we might expect: larger networks may suffer more from overfitting and thus regularization can help more. As an example, $L_2$ sometimes helps more with thinner network (TRPO Ant), and sometimes more with wider network (PPO HumanoidStandup).

## 6 POLICY AND VALUE NETWORK REGULARIZATION

Our experiments in previous sections only impose regularization on the policy network. To investigate the relationship between policy and value network regularization, we evaluate four options: 1) no regularization, and regularizing 2) policy network, 3) value network, 4) policy and value networks. For 2) and 3) we tune the regularization strengths independently and then use the appropriate ones for 4) (more details in Appendix B). We evaluate all four algorithms on the six MuJoCo tasks and present the percentage of tasks where we obtain improvement in Table 6. Note that entropy regularization is not applicable to the value network. For detailed training curves, please refer to Appendix L.

| Reg\Alg | A2C | | | TRPO | | | PPO | | | SAC | | | TOTAL | | |
|---|---|---|---|---|---|---|---|---|---|---|---|---|---|---|---|
| | Pol | Val | P+V | Pol | Val | P+V | Pol | Val | P+V | Pol | Val | P+V | Pol | Val | P+V |
| $L_2$ | **50.0** | 0.0 | 16.7 | **50.0** | 16.7 | 33.3 | **66.7** | 16.7 | **66.7** | **66.7** | 33.3 | 33.3 | **58.3** | 16.7 | 37.5 |
| $L_1$ | **50.0** | 16.7 | **50.0** | 33.3 | 0.0 | **33.3** | **66.7** | 0.0 | 50.0 | **33.3** | 33.3 | 33.3 | **45.8** | 12.5 | 41.7 |
| Weight Clip | 16.7 | 0.0 | **16.7** | **50.0** | 33.3 | 16.7 | **66.7** | 0.0 | **66.7** | **33.3** | 16.7 | 16.7 | **41.7** | 8.3 | 29.2 |
| Dropout | 0.0 | **16.7** | 0.0 | N/A | **33.3** | N/A | **66.7** | 33.3 | 50.0 | **50.0** | 0.0 | 0.0 | **38.9** | 20.8 | 16.7 |
| BatchNorm | **16.7** | 16.7 | 16.7 | 0.0 | **16.7** | 0.0 | 16.7 | 0.0 | **50.0** | **33.3** | 16.7 | 0.0 | **16.7** | 12.5 | **16.7** |

Table 6: Percentage (%) of environments where the final performance "improves" when applying regularization on policy / value / policy and value networks.

It can be seen that generally, only regularizing the policy network tends to be the most effective for almost all algorithms and regularizations. Regularizing the value network alone does not bring as significant performance improvement as regularizing the policy network alone. Though regularizing both is better than regularizing value network alone, it is still worse than only regularizing the policy network.

## 7 DISCUSSION AND CONCLUSION

**Why does regularization benefit policy optimization?** In RL, we are typically training and evaluating on the same environment, i.e., there is no generalization gap across different environments. However, there is still generalization between samples: the agents is only trained on the limited trajectories it has experienced, which cannot cover the whole state-action space of the environment. A successful policy needs to generalize from seen samples to unseen ones, which potentially makes regularization necessary in RL. This might also explain why regularization could be more helpful on harder tasks, which have larger state space. In this case, the portion of the space that have appeared in training tends to be smaller, and overfitting to this smaller portion of space would cause more serious issues, in which case regularizations may help. Some detailed analysis are provided in Appendix G.

**Why do BN and dropout work only with off-policy algorithms?** One major finding in our experiments is BN and dropout can sometimes improve on the off-policy algorithm SAC, but mostly would hurt on-policy algorithms. There are two possible reasons for this: 1) for both BN and dropout, training mode is used to train the network, and testing mode is used to sample actions during interaction with the environment, leading to a discrepancy between the sampling policy and optimization policy (the same holds if we always use training mode). For on-policy algorithms, if such discrepancy is large, it can cause severe off-policy issues, which hurts the optimization process or even crashes it. For off-policy algorithms, this discrepancy is not an issue since they naturally accept off-policy data. 2) Batch Normalization layers can be sensitive to input distribution shifts, since the mean and std statistics depend heavily on the input, and if the input distribution changes too quickly in training, the mapping functions of BN layers can change quickly too, and it can possibly destabilize training. One evidence for this is that in supervised learning, when transferring a ImageNet pretrained model to other vision datasets, sometimes the BN layers are fixed (Yang et al., 2017) and only other layers are trained. In on-policy algorithms, we always use the samples generated from the latest policy; in off-policy algorithms, the sample distributions are relatively slow-changing since we always draw from the whole replay buffer which holds cumulative data. The faster-changing input distribution for on-policy algorithms could be harmful to BN. Previously, BN has also been shown to be effective in Deep Deterministic Policy Gradient (DDPG) (Lillicrap et al., 2015), an off-policy algorithm.

**In summary**, we conducted the first comprehensive study of regularization methods with multiple policy optimization algorithms on continuous control benchmarks. We found that $L_2$ regularization,

despite being largely ignored in prior literature, is effective in improving performance, even more than the widely used entropy regularization. BN and dropout could also be useful but only on off-policy algorithms. Our findings were confirmed with multiple hyperparameters. Further experiments have shown that generally the best practice is to regularize the policy network alone but not the value network or both.

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

# APPENDIX

## A  POLICY OPTIMIZATION ALGORITHMS

The policy optimization family of algorithms is one of the most popular methods for solving reinforcement learning problems. It directly parameterizes and optimizes the policy to gain more cumulative rewards. Below, we give a brief introduction to the algorithms we evaluate in our work.

**A2C.** Sutton et al. (2000) developed a policy gradient to update the parametric policy in a gradient descent manner. However, the gradient estimated in this way suffers from high variance. Advantage Actor Critic (A3C) (Mnih et al., 2016) is proposed to alleviate this problem by introducing a function approximator for values and replacing the Q-values with advantage values. A3C also utilizes multiple actors to parallelize training. The only difference between A2C and A3C is that in a single training iteration, A2C waits for parallel actors to finish sampling trajectories before updating the neural network parameters, while A3C updates in an asynchronous manner.

**TRPO.** Trust Region Policy Optimization (TRPO) (Schulman et al., 2015) proposes to constrain each update within a safe region defined by KL divergence to guarantee policy improvement during training. Though TRPO is promising at obtaining reliable performance, approximating the KL constraint is quite computationally heavy.

**PPO.** Proximal Policy Optimization (PPO) (Schulman et al., 2017) simplifies TRPO and improves computational efficiency by developing a surrogate objective that involves clipping the probability ratio to a reliable region, so that the objective can be optimized using first-order methods.

**SAC.** Soft Actor Critic (SAC) (Haarnoja et al., 2018) optimizes the maximum entropy objective in reward (Ziebart et al., 2008), which is different from the objective of the on-policy methods above. SAC combines soft policy iteration, which maximizes the maximum entropy objective, and clipped double $Q$ learning (Fujimoto et al., 2018), which prevents overestimation bias, during actor and critic updates, respectively.

## B  IMPLEMENTATION AND TUNING FOR REGULARIZATION METHODS

As mentioned in the paper, in Section 4 we only regularize the policy network; in Section 6, we investigate regularizing both policy and value networks.

For $L_1$ and $L_2$ regularization, we add $\lambda||\theta||_1$ and $\frac{\lambda}{2}||\theta||_2^2$, respectively, to the loss of policy network or value network of each algorithm (for SAC's value regularization, we apply regularization only to the $V$ network instead of also to the two $Q$ networks). $L_1$ and $L_2$ loss are applied to all the weights of the policy or value network. For A2C, TRPO, and PPO, we tune $\lambda$ in the range of $[1e-5, 5e-5, 1e-4, 5e-4]$ for $L_1$ and $[5e-5, 1e-4, 5e-4, 1e-3]$ for $L_2$. For SAC, we tune $\lambda$ in the range of $[5e-4, 1e-3, 5e-3, 1e-2]$ for $L_1$ and $[1e-3, 5e-3, 1e-2, 5e-2]$ for $L_2$.

For weight clipping, the OpenAI Baseline implementation of the policy network of A2C, TRPO, and PPO outputs the mean of policy action from a two-layer fully connected network (MLP). The log standard deviation of the policy action is represented by a standalone trainable vector. We find that when applied only to the weights of MLP, weight clipping makes the performance much better than when applied to only the logstd vector or both. Thus, for these three algorithms, the policy network weight clipping results shown in all the sections above come from clipping only the MLP part of the policy network. On the other hand, in the SAC implementation, both the mean and the log standard deviation come from the same MLP, and there is no standalone log standard deviation vector. Thus, we apply weight clipping to all the weights of the MLP. For all algorithms, we tune the policy network clipping range in $[0.1, 0.2, 0.3, 0.5]$. For value network, the MLP produces a single output of estimated value given a state, so we clip all the weights of the MLP. For A2C, TRPO, and PPO, we tune the clipping range in $[0.1, 0.2, 0.3, 0.5]$. For SAC, we only clip the $V$ network and do not clip the two $Q$ networks for simplicity. We tune the clipping range in $[0.3, 0.5, 0.8, 1.0]$ due to its weights having larger magnitude.

For Batch Normalization/dropout, we apply it before the activation function of each hidden layer/immediately after the activation function. When the policy or the value network is performing update using minibatches of trajectory data or minibatches of replay buffer data, we use the train mode of regularization and update the running mean and standard deviation. When the policy is sampling trajectory from the environment, we use the test mode of regularization and use the existing running mean and standard deviation to normalize data. For Batch Normalization/dropout on value network, only training mode is applied since value network does not participate in sampling trajectories. Note that adding policy network dropout on TRPO causes the KL divergence constraint $\mathbb{E}_{s \sim \rho_{\theta_{\text{old}}}}\left[D_{\text{KL}}\left(\pi_{\theta_{\text{old}}}(\cdot|s)\|\pi_{\theta}(\cdot|s)\right)\right] \leq \delta$ to be violated almost every time during policy network update. Thus, policy network dropout causes the training to fail on TRPO, as the policy network cannot be updated.

For entropy regularization, we add $-\lambda L^H$ to the policy loss. $\lambda$ is tuned from $[5e-5, 1e-4, 5e-4, 1e-3]$ for A2C, TRPO, PPO and $[0.1, 0.5, 1.0, 5.0]$ for SAC. Note that for SAC, our entropy regularization is added directly on the optimization objective (equation 12 in Haarnoja et al. (2018)), and is different from the original maximum entropy objective inside the reward term.

The optimal policy network regularization strength we selected for each algorithm and environment used in Section 4 can be seen in the legends of Appendix L. In addition to the environment-specific-strength regularization results presented in Section 4, we also present the results when the regularization strength is fixed across all environments for the same algorithm. The results are shown in Appendix H.

In Section 6, to investigate the effect of regularizing both policy and value networks, we combine the tuned optimal policy and value network regularization strengths. The detailed training curves are presented in Appendix L.

As a side note, when training A2C, TRPO, and PPO on the HalfCheetah environment, the results have very large variance. Thus, for each regularization method, after we obtain the best strength, we rerun it for another five seeds as the final result in Table 1 and 2.

## C  ADDITIONAL TRAINING CURVES

As a complement with Figure 1 in Section 4, we plot the training curves of the other five environments in Figure 4.

## D  DEFAULT HYPERPARAMETER SETTINGS FOR BASELINES

**Training timesteps.** For A2C, TRPO, and PPO, we run $5e6$ timesteps for Hopper, Walker, and HalfCheetah; $2e7$ timesteps for Ant, Humanoid (MuJoCo), and HumanoidStandup; $5e7$ timesteps for Humanoid (RoboSchool); and $1e8$ timesteps for AtlasForwardWalk and HumanoidFlagrun. For SAC, since its simulation speed is much slower than A2C, TRPO, and PPO (as SAC updates its policy and value networks using a minibatch of replay buffer data at every timestep), and since it takes fewer timesteps to converge, we run $1e6$ timesteps for Hopper and Walker; $3e6$ timesteps for HalfCheetah and Ant; $5e6$ timesteps for Humanoid and HumanoidStandup; and $1e7$ timesteps for the RoboSchool environments.

**Hyperparameters for RoboSchool.** In the original PPO paper (Schulman et al., 2017), hyperparameters for the Roboschool tasks are given, so we apply the same hyperparameters to our training, except that instead of linear annealing the log standard deviation of action distribution from $-0.7$ to $-1.6$, we let it to be learnt by the algorithm, as implemented in OpenAI Baseline (Dhariwal et al., 2017). For TRPO, due to its proximity to PPO, we copy PPO's hyperparameters if they exist in both algorithms. We then tune the value update step size in $[3e-4, 5e-4, 1e-3]$. For A2C, we keep the original hyperparameters and tune the number of actors in $[32, 128]$ and the number of timesteps for each actor between consecutive policy updates in $[5, 16, 32]$. For SAC, we tune the reward scale from $[5, 20, 100]$.

The detailed hyperparameters used in our baselines for both MuJoCo and RoboSchool are listed in Tables 7-10.

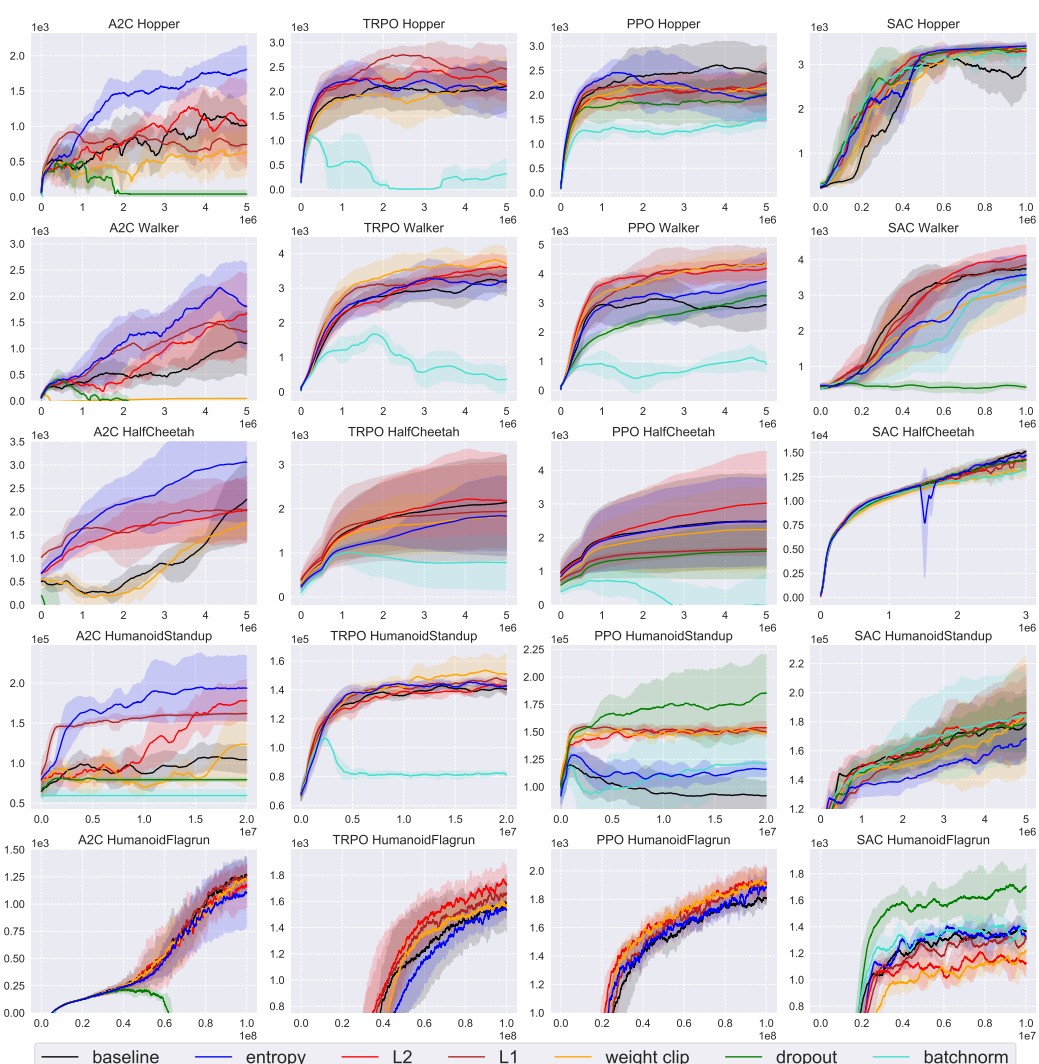

Figure 4: Reward vs. timesteps, for four algorithms (columns) and three environments (rows).

| Hyperparameter | Value |
|---|---|
| Hidden layer size | $64 \times 2$ |
| Sharing policy and value weights | False |
| Number of hidden layers | 2 |
| Rollout timesteps per actor | 5 |
| Number of actors | 1 |
| Step size | $7e - 4$, linear decay |
| Max gradient norm | 0.5 |
| Discount factor ($\gamma$) | 0.99 |

| Hyperparameter | Value |
|---|---|
| Hidden layer size | $64 \times 2$ |
| Number of hidden layers | 2 |
| Sharing policy and value weights | False |
| Rollout timesteps per actor | 32 |
| Number of actors | 32 (Humanoid, Atlas) 128 (Flagrun) |
| Step size | $7e - 4$, linear decay |
| Max gradient norm | 0.5 |
| Discount factor ($\gamma$) | 0.99 |

Table 7: Hyperparameter setting for A2C MuJoCo and RoboSchool tasks

| Hyperparameter | Value |
| --- | --- |
| Hidden layer size | $32 \times 2$ |
| Number of hidden layers | 2 |
| Sharing policy and value weights | False |
| Rollout timesteps per actor | 1024 |
| Number of actors | 1 |
| Value network step size | $1e-3$, constant |
| Max KL divergence | 0.01 |
| Discount factor ($\gamma$) | 0.99 |
| GAE parameter ($\lambda$) | 0.98 |
| Conjugate gradient damping | 0.1 |
| Conjugate gradient iterations | 10 |
| Value network optimization epochs | 10 |
| Value network update minibatch size | 64 |
| Probability ratio clipping range | 0.2 |

| Hyperparameter | Value |
| --- | --- |
| Hidden layer size | $64 \times 2$ |
| Number of hidden layers | 2 |
| Sharing policy and value weights | False |
| Rollout timesteps per actor | 512 |
| Number of actors | 32 (Humanoid, Atlas) 128 (Flagrun) |
| value network step size | $1e-3$, constant |
| Max KL divergence | 0.01 |
| Discount factor ($\gamma$) | 0.99 |
| GAE parameter ($\lambda$) | 0.98 |
| Conjugate gradient damping | 0.1 |
| Conjugate gradient iterations | 10 |
| Value network optimization epochs | 15 |
| Value network update minibatch size | 4096 |
| Probability ratio clipping range | 0.2 |

Table 8: Hyperparameter setting for TRPO Mujoco and RoboSchool tasks. The original OpenAI implementation does not support multiple actors sampling trajectories at the same time, so we modified the code to support this feature and accelerate training.

| Hyperparameter | Value |
| --- | --- |
| Hidden layer size | $64 \times 2$ |
| Number of hidden layers | 2 |
| Sharing policy and value weights | False |
| Rollout timesteps per actor | 2048 |
| Number of actors | 1 |
| Number of minibatches | 32 |
| Step size | $3e-4$, linear decay |
| Max gradient norm | 0.5 |
| Discount factor ($\gamma$) | 0.99 |
| GAE parameter ($\lambda$) | 0.95 |
| Number of optimization epochs | 10 |
| Probability ratio clipping range | 0.2 |

| Hyperparameter | Value |
| --- | --- |
| Hidden layer size | $64 \times 2$ |
| Number of hidden layers | 2 |
| Sharing policy and value weights | False |
| Rollout timesteps per actor | 512 |
| Number of actors | 32 (Humanoid, Atlas) 128 (Flagrun) |
| Minibatch size | 4096 |
| Step size | $3e-4$, linear decay |
| Max gradient norm | 0.5 |
| Discount factor ($\gamma$) | 0.99 |
| GAE parameter ($\lambda$) | 0.95 |
| Number of optimization epochs | 15 |
| Probability ratio clipping range | 0.2 |

Table 9: Hyperparameter setting for PPO MuJoCo and RoboSchool tasks

| Hyperparameter | Value |
| --- | --- |
| Hidden layer size | $256 \times 2$ |
| Number of hidden layers | 2 |
| Samples per batch | 256 |
| Replay buffer size | $10^6$ |
| Learning rate | $3e-4$ constant |
| Discount factor ($\gamma$) | 0.99 |
| Target smoothing coefficient ($\tau$) | 0.005 |
| Target update interval | 1 |
| Reward Scaling | 5 (Hopper, Walker, HalfCheetah, Ant) 20 (MuJoCo Humanoid and all RoboSchool tasks) 100 (HumanoidStandup) |

Table 10: Hyperparameter setting for SAC

## E  HYPERPARAMETER SAMPLING DETAILS

In Section 5, we present results based on five hyperparameter settings. To obtain such hyperparameter variations, we consider varying the learning rates and the hyperparameters that each algorithm is very sensitive to. For A2C, TRPO, and PPO, we consider a range of rollout timesteps between consecutive policy updates by varying the number of actors or the number of trajectory sampling timesteps for each actor. For SAC, we consider a range of reward scale and a range of target smoothing coefficient.

More concretely, for A2C, we sample the learning rate from $[2e-4, 7e-4, 2e-3]$ linear decay, the number of trajectory sampling timesteps (nsteps) for each actor from $[3, 5, 16, 32]$, and the number of actors (nenvs) from $[1, 4]$. For TRPO, we sample the learning rate of value network (vf_stepsize) from $[3e-4, 5e-4, 1e-3]$ and the number of trajectory sampling timesteps for each actor (nsteps) in $[1024, 2048, 4096, 8192]$. The policy update uses conjugate gradient descent and is controlled by the max KL divergence. For PPO, we sample the learning rate from $[1e-4$ linear, $3e-4$ constant$]$, the number of actors (nenvs) from $[1, 2, 4, 8]$, and the probability ratio clipping range (cliprange) in $[0.1, 0, 2]$. For SAC, we sample the learning rate from $[1e-4, 3e-4, 1e-3]$ the target smoothing coefficient ($\tau$) from $[0.001, 0.005, 0.01]$, and the reward scale from small, default, and large mode.

The default reward scale of 5 is changed to $(3, 5, 20)$; 20 to $(4, 20, 100)$; 100 to $(20, 100, 400)$ for each mode, respectively. Sampled hyperparameters 1-5 for each algorithms are listed in Table 11a-11d.

| | Learning rate | Nsteps | Nenvs |
|---|---|---|---|
| Baseline | $7e-4$ | 5 | 1 |
| Hyperparam. 1 | $2e-3$ | 32 | 4 |
| Hyperparam. 2 | $2e-3$ | 32 | 1 |
| Hyperparam. 3 | $7e-4$ | 16 | 1 |
| Hyperparam. 4 | $7e-4$ | 32 | 4 |
| Hyperparam. 5 | $2e-4$ | 3 | 4 |

(a) Sampled hyperparameter settings for A2C

| | Vf_stepsize | Nsteps |
|---|---|---|
| Baseline | $1e-3$ | 1024 |
| Hyperparam. 1 | $5e-4$ | 8192 |
| Hyperparam. 2 | $1e-3$ | 4096 |
| Hyperparam. 3 | $3e-4$ | 2048 |
| Hyperparam. 4 | $5e-4$ | 1024 |
| Hyperparam. 5 | $5e-4$ | 4096 |

(b) Sampled hyperparameter settings for TRPO

| | Learning rate | Nenvs | Cliprange |
|---|---|---|---|
| Baseline | $3e-4$ linear | 1 | 0.2 |
| Hyperparam. 1 | $3e-4$ linear | 8 | 0.2 |
| Hyperparam. 2 | $1e-4$ constant | 8 | 0.2 |
| Hyperparam. 3 | $3e-4$ linear | 4 | 0.1 |
| Hyperparam. 4 | $1e-4$ constant | 2 | 0.2 |
| Hyperparam. 5 | $3e-4$ linear | 1 | 0.1 |

(c) Sampled hyperparameter settings for PPO

| | Learning rate | $\tau$ | Mode |
|---|---|---|---|
| Baseline | $3e-4$ | 0.005 | default |
| Hyperparam. 1 | $3e-4$ | 0.005 | small |
| Hyperparam. 2 | $1e-4$ | 0.001 | large |
| Hyperparam. 3 | $1e-3$ | 0.005 | small |
| Hyperparam. 4 | $3e-4$ | 0.01 | small |
| Hyperparam. 5 | $1e-3$ | 0.005 | default |

(d) Sampled hyperparameter settings for SAC

Table 11: Sampled hyperparameter settings for Section 5

## F    HYPERPARAMETER EXPERIMENT IMPROVEMENT PERCENTAGE

We provide the percentage of improvement result in Table 12 as a complement with Table 4.

| Reg \ Alg | A2C | | | TRPO | | | PPO | | | SAC | | | TOTAL | | |
|---|---|---|---|---|---|---|---|---|---|---|---|---|---|---|---|
| | Easy | Hard | Total | Easy | Hard | Total | Easy | Hard | Total | Easy | Hard | Total | Easy | Hard | Total |
| Entropy | **20.0** | 40.0 | 32.0 | 0.0 | 26.7 | 16.0 | 10.0 | 33.3 | 24.0 | **60.0** | 13.3 | **32.0** | **22.5** | 28.3 | 26.0 |
| $L_2$ | **20.0** | **60.0** | **44.0** | 10.0 | 40.0 | 28.0 | **20.0** | **86.7** | **60.0** | 10.0 | **40.0** | 28.0 | 15.0 | **56.7** | **40.0** |
| $L_1$ | 10.0 | 53.3 | 36.0 | 10.0 | **46.7** | 32.0 | 10.0 | **86.7** | 56.0 | 20.0 | 26.7 | 24.0 | 12.5 | 53.3 | 37.0 |
| Weight Clip | 0.0 | 46.7 | 28.0 | **40.0** | **46.7** | **44.0** | 10.0 | 73.3 | 48.0 | 0.0 | 33.3 | 20.0 | 12.5 | 50.0 | 35.0 |
| Dropout | **20.0** | 0.0 | 8.0 | N/A | N/A | N/A | 0.0 | 40.0 | 24.0 | 0.0 | 20.0 | 12.0 | 6.7 | 20.0 | 14.7 |
| BatchNorm | 0.0 | 0.0 | 0.0 | 10.0 | 0.0 | 4.0 | 10.0 | 33.3 | 24.0 | 20.0 | 20.0 | 20.0 | 10.0 | 13.3 | 12.0 |

Table 12: Percentage (%) of environments where the final performance "improves" when using regularization, under five randomly sampled training hyperparameters for each algorithm.

## G    GENERALIZATION ANALYSIS

We provide some experiments to accompany the qualitative analysis on generalization in section 7.

We investigate the agent's obtained rewards on a set of sampled trajectories. We train PPO Humanoid and TRPO Ant models, then evaluate the reward on 100 trajectories and plot the reward distribution in Figure 5 and 6. These trajectories are unseen samples during training, since the state space is continuous. The trajectory reward's distributions are plotted in the figures. For baseline, some of the trajectories yield relatively high rewards, while others yield low rewards, demonstrating the baseline cannot stably generalize to unseen examples; for regularized models, the rewards are mostly high and have smaller variance, demonstrating they can more stably generalize to unseen samples. This suggests that conventional regularization can improve the model's generalization ability to larger portion of unseen samples.

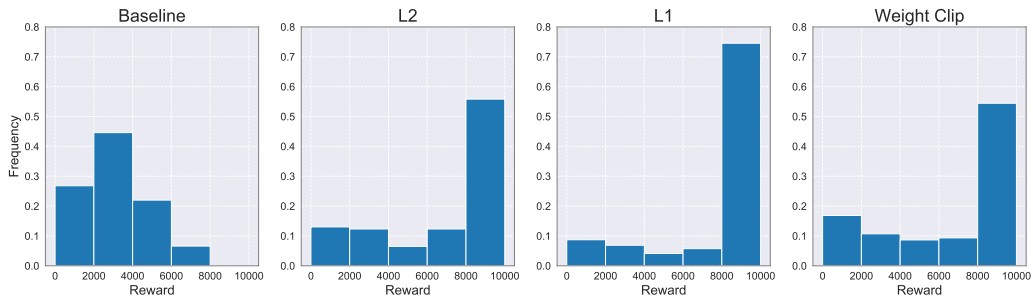

Figure 5: The distribution of reward over 100 trajectories for PPO Humanoid after running $2e7$ timesteps. The bins indicate reward range and the y-axis indicates the frequency of trajectory rewards in a range.

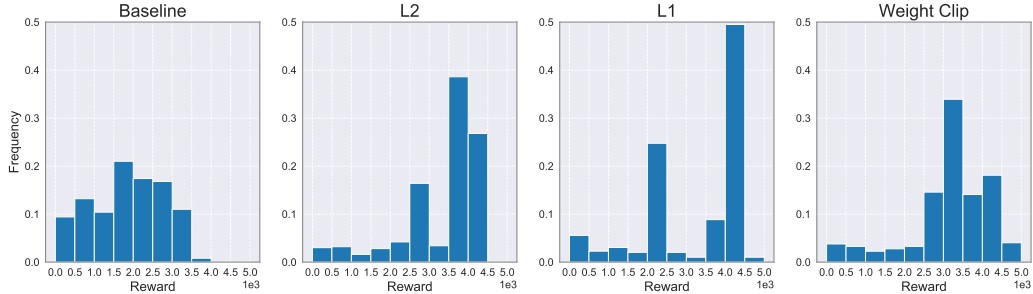

Figure 6: The distribution of reward over 100 trajectories for TRPO Ant after running $5e6$ timesteps.

Next, we present the results of varying the number of training samples/timesteps in Figure 7. We find that for regularized models to reach the same level of reward as baseline, they only need much fewer samples in training. Note that the reward is also on unseen samples/trajectories. In addition, regularization's gain over baseline can be larger when the samples are fewer (SAC Ant, TRPO Ant). Since the ability to learn from fewer samples is closely related to the notion of generalization, we can conclude that regularized models have better generalization ability than baselines.

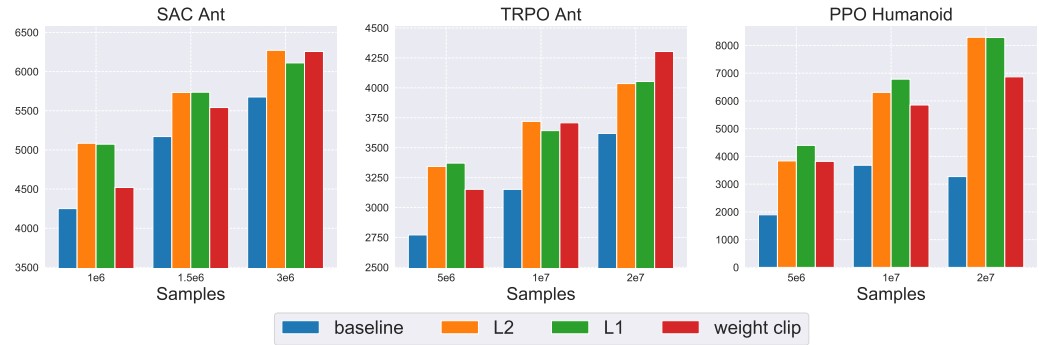

Figure 7: The performance comparison between regularization and baseline after training for various amounts of timesteps. Models under conventional regularization achieve the same performance with less timesteps than the baseline.

## H  REGULARIZATION WITH A SINGLE STRENGTH

In previous sections, we tune the strength of regularization for each algorithm and environment, as described in Appendix B. Now we restrict the regularization methods to a **single** strength for each algorithm, across different environments. The results are shown in Table 13, 14, and 15. The selected strength are presented in Table 16. We see that the $L_2$ regularization is still generally the best performing one, but SAC is an exception, where BN is better. This can be explained by the fact that in SAC, the reward scaling coefficient is different for each environment, which potentially causes the optimal $L_2$ and $L_1$ strength to vary a lot across different environments, while BN does not have a strength parameter.

| Reg \ Alg | A2C | | | TRPO | | | PPO | | | SAC | | | TOTAL | | |
|---|---|---|---|---|---|---|---|---|---|---|---|---|---|---|---|
| | Easy | Hard | Total | Easy | Hard | Total | Easy | Hard | Total | Easy | Hard | Total | Easy | Hard | Total |
| Entropy | **33.3** | **66.7** | **55.6** | 0.0 | 33.3 | 22.2 | 0.0 | 16.7 | 11.1 | 0.0 | 16.7 | 11.1 | 8.3 | 33.3 | 25.0 |
| $L_2$ | 0.0 | 50.0 | 33.3 | 0.0 | **50.0** | **33.3** | **33.3** | **66.7** | **55.6** | 33.3 | 33.3 | 33.3 | 16.7 | **50.0** | **38.9** |
| $L_1$ | 0.0 | 33.3 | 22.2 | 0.0 | **50.0** | **33.3** | **33.3** | 50.0 | 44.4 | 33.3 | 33.3 | 33.3 | 16.7 | 41.7 | 33.3 |
| Weight clipping | 0.0 | 0.0 | 0.0 | **33.3** | 33.3 | **33.3** | **33.3** | 50.0 | 44.4 | 33.3 | 0.0 | 11.1 | 25.0 | 20.8 | 22.2 |
| Dropout | 0.0 | 0.0 | 0.0 | N/A | N/A | N/A | **33.3** | 50.0 | 44.4 | **66.7** | 16.7 | 33.3 | **33.3** | 22.2 | 25.9 |
| BatchNorm | 0.0 | 0.0 | 0.0 | 0.0 | 0.0 | 0.0 | 0.0 | 16.7 | 11.1 | 33.3 | **50.0** | **44.4** | 8.3 | 16.7 | 13.9 |

Table 13: Percentage (%) of environments that, when using a regularization, "improves". For each algorithm, one single strength for each regularization is applied to all environments.

| Reg \ Alg | A2C | | | TRPO | | | PPO | | | SAC | | | TOTAL | | |
|---|---|---|---|---|---|---|---|---|---|---|---|---|---|---|---|
| | Easy | Hard | Total | Easy | Hard | Total | Easy | Hard | Total | Easy | Hard | Total | Easy | Hard | Total |
| Entropy | **1.67** | 2.17 | **2.00** | 3.33 | 2.83 | 3.00 | 4.00 | 4.50 | 4.33 | 5.00 | 4.50 | 4.67 | 3.50 | 3.50 | 3.50 |
| $L_2$ | 2.00 | **2.00** | **2.00** | 3.33 | 2.50 | **2.78** | 3.00 | **2.33** | **2.55** | 4.67 | 4.00 | 4.22 | **3.25** | **2.71** | **2.89** |
| $L_1$ | 3.33 | 3.83 | 3.66 | 4.00 | **2.17** | **2.78** | 3.00 | 3.33 | 3.22 | 4.33 | 4.00 | 4.11 | 3.67 | 3.33 | 3.44 |
| Weight clipping | 5.00 | 3.33 | 3.89 | **2.00** | 3.83 | 3.22 | **2.67** | 3.17 | 3.00 | 4.00 | 5.50 | 5.00 | 3.42 | 3.96 | 3.78 |
| Dropout | 6.00 | 6.00 | 6.00 | N/A | N/A | N/A | 5.33 | 4.17 | 4.56 | **2.00** | 3.50 | **3.00** | 4.44 | 4.56 | 4.52 |
| BatchNorm | 7.00 | 7.00 | 7.00 | 6.00 | 6.00 | 6.00 | 7.00 | 5.33 | 5.89 | 4.33 | **2.33** | **3.00** | 6.08 | 5.17 | 5.47 |

Table 14: The average rank in the mean return for different regularization methods. For each algorithm, one single strength for each regularization is applied to all environments.

| Reg \ Alg | A2C | | | TRPO | | | PPO | | | SAC | | | TOTAL | | |
|---|---|---|---|---|---|---|---|---|---|---|---|---|---|---|---|
| | Easy | Hard | Total | Easy | Hard | Total | Easy | Hard | Total | Easy | Hard | Total | Easy | Hard | Total |
| Entropy | 0.00 | 0.50 | 0.47 | 0.47 | 1.15 | 1.26 | 0.82 | 2.03 | 1.81 | 2.16 | 2.11 | 2.17 | 1.75 | 1.90 | 1.86 |
| $L_2$ | 0.94 | 0.50 | 0.87 | 0.47 | 1.07 | 1.15 | 1.63 | 0.69 | 1.17 | 0.82 | 1.70 | 1.47 | 1.26 | 1.21 | 1.23 |
| $L_1$ | 0.94 | 0.75 | 0.87 | 0.47 | 1.07 | 0.94 | 1.70 | 0.94 | 1.29 | 0.94 | 1.21 | 1.26 | 1.26 | 1.43 | 1.38 |
| Weight clipping | 0.47 | 0.69 | 0.63 | 1.41 | 1.37 | 1.42 | 0.82 | 1.26 | 1.15 | 2.05 | 1.34 | 1.62 | 1.44 | 1.44 | 1.45 |
| Dropout | 0.00 | 0.00 | 0.00 | N/A | N/A | N/A | 0.47 | 1.97 | 1.70 | 1.70 | 2.11 | 1.99 | 1.61 | 2.04 | 1.91 |
| BatchNorm | 0.00 | 0.00 | 0.00 | 0.00 | 0.00 | 0.00 | 0.00 | 0.90 | 0.92 | 0.47 | 1.71 | 1.95 | 0.49 | 1.61 | 1.42 |

Table 15: The standard deviation of rank in the mean return for different regularization methods, when one single strength for each regularization is applied to all environments.

| Reg \ Alg | A2C | TRPO | PPO | SAC |
|---|---|---|---|---|
| Entropy | $5e-4$ | $5e-4$ | $5e-4$ | 1.0 |
| $L_2$ | $1e-4$ | $5e-4$ | $5e-4$ | $5e-2$ |
| $L_1$ | $1e-4$ | $1e-4$ | $1e-4$ | $5e-3$ |
| Weight clipping | 0.2 | 0.2 | 0.2 | 0.3 |
| Dropout | 0.05 | 0.05 | 0.05 | 0.2 |
| BatchNorm | True | True | True | True |

Table 16: The fixed single regularization strengths that are used in each algorithm to obtain results in Table 13 and Table 14.

## I  REGULARIZING WITH BOTH $L_2$ AND ENTROPY

We also investigate the effect of combining $L_2$ regularization with entropy regularization, given that both cases of applying one of them alone yield performance improvement. We take the optimal strength of $L_2$ regularization and entropy regularization together and compare with applying $L_2$ regularization or entropy regularization alone. We find that the performance increases for PPO HumanoidStandup, approximately stays the same for TRPO Ant, and decreases for A2C Humanoid-Standup. Thus, the regularization benefits are not always addable. This phenomenon is possibly caused by the fact that the algorithms already achieve good performance using only $L_2$ regularization or entropy regularization, and further performance improvement is restrained by the intrinsic capabilities of algorithms.

## J  COMPARING $L_2$ REGULARIZATION WITH FIXED WEIGHT DECAY (ADAMW)

For the Adam optimizer (Kingma & Ba, 2015), "fixed weight decay" (AdamW in Loshchilov & Hutter (2019)) differs from $L_2$ regularization in that the gradient of $\frac{1}{2}\lambda||\theta||^2$ is not computed with the gradient of the original loss, but the weight is "decayed" finally with the gradient update. For Adam these two procedures are very different (see Loshchilov & Hutter (2019) for more details). In this section, we compare the effect of adding $L_2$ regularization with that of using AdamW, with PPO on Humanoid and HumanoidStandup. The result is shown in Figure 9. Similar to $L_2$, we briefly tune the strength of weight decay in AdamW and the optimal one is used. We find that while both $L_2$ regularization and AdamW can significantly improve the performance over baseline, the performance of AdamW tends to be slightly lower than the performance of $L_2$ regularization.

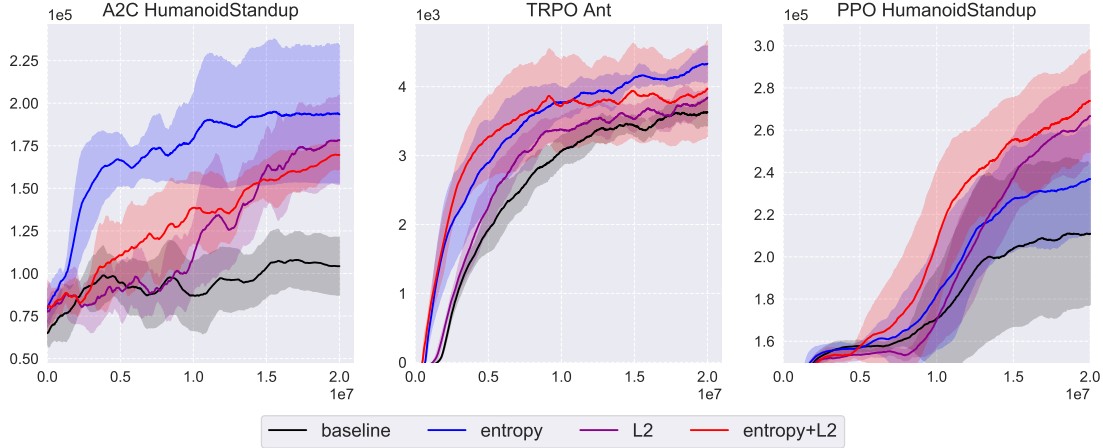

Figure 8: The effect of combining $L_2$ regularization with entropy regularization. For PPO Humanoid-Standup, we use the third randomly sampled hyperparameter setting. For A2C HumanoidStandup and TRPO Ant, we use the baseline as in Section 4.

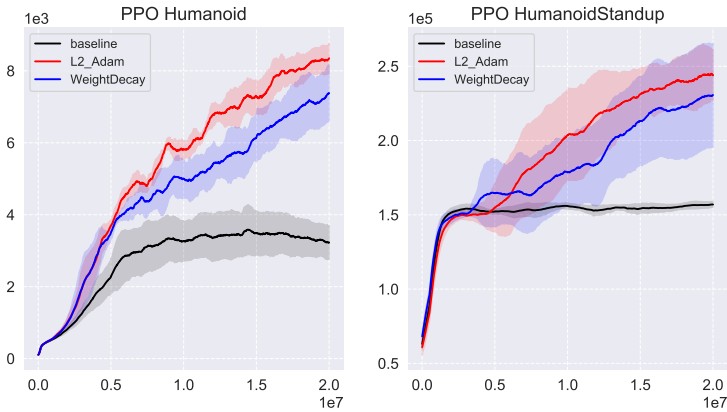

Figure 9: Comparison between $L_2$ regularization and weight decay. For PPO Humanoid and HumanoidStandup, we use the third randomly sampled hyperparameter setting.

## K   TRAINING CURVES FOR HYPERPARAMETER EXPERIMENTS

In this section, we plot the full training curves of the experiments in Section 5 with five sampled hyperparameter settings for each algorithm from Figure 10 to Figure 13. The strength of each regularization is tuned according to the range in Appendix B.

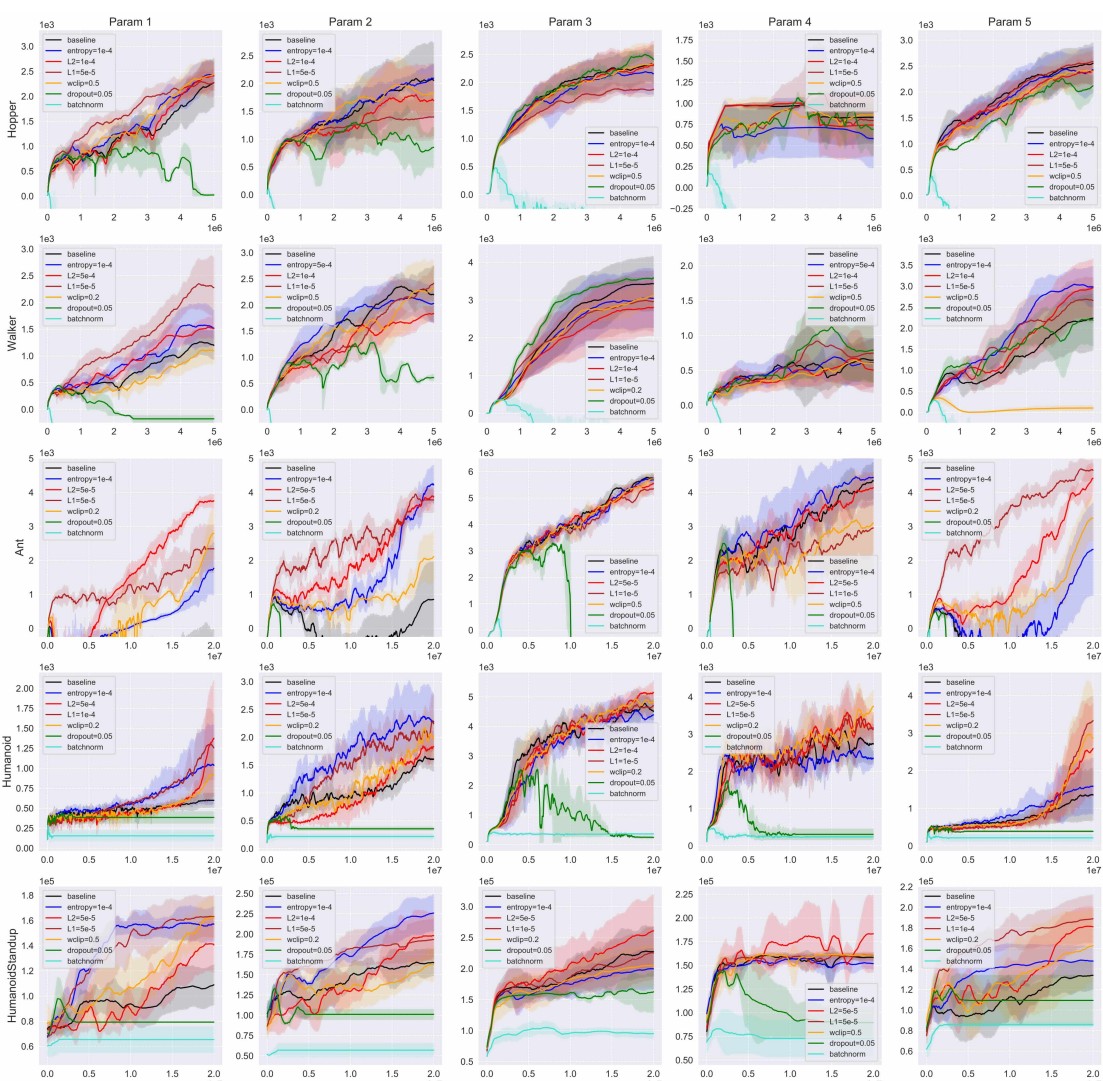

Figure 10: Training curves of A2C regularizations under five randomly sampled hyperparameters.

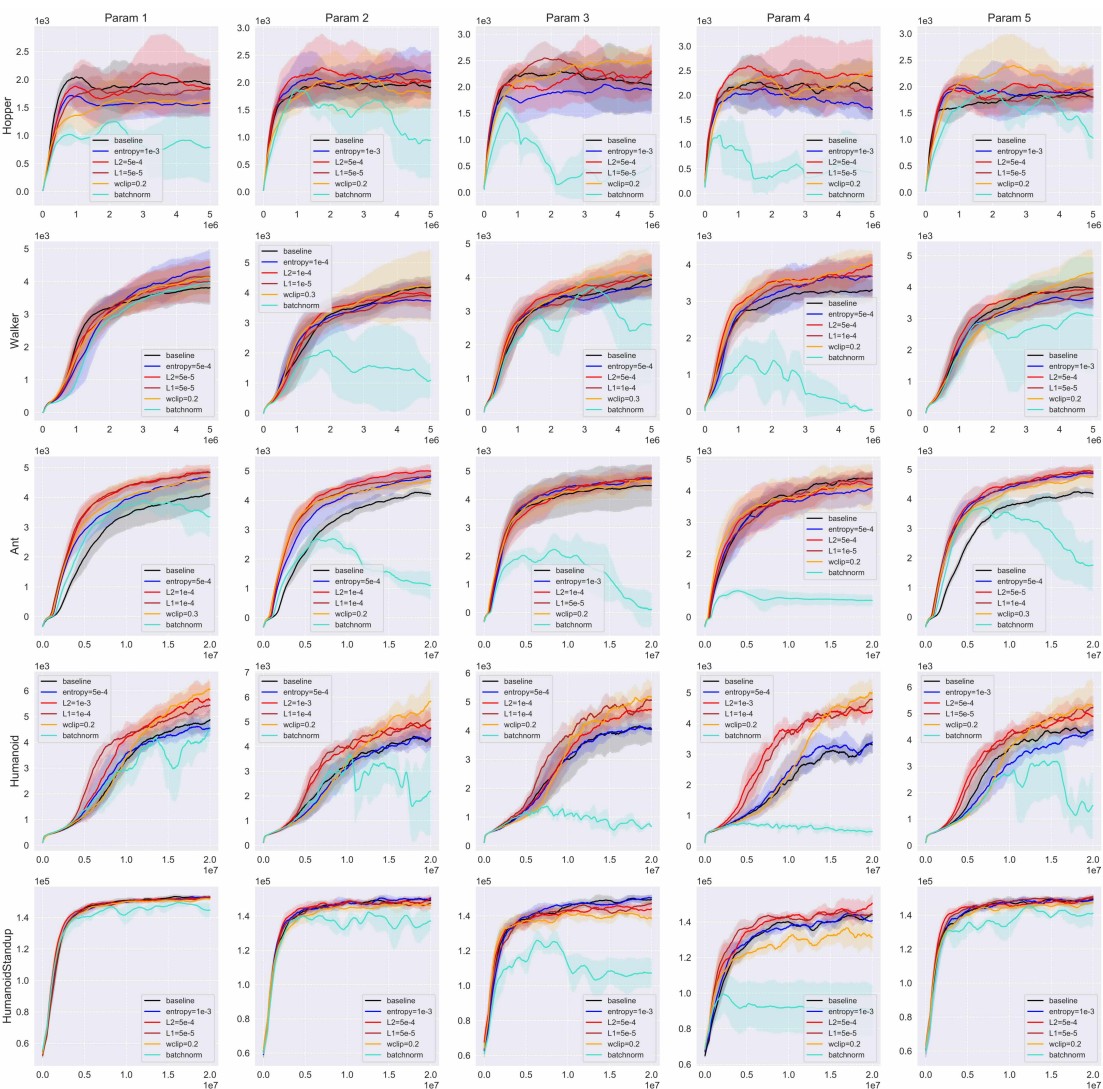

Figure 11: Training curves of TRPO regularizations under five randomly sampled hyperparameters.

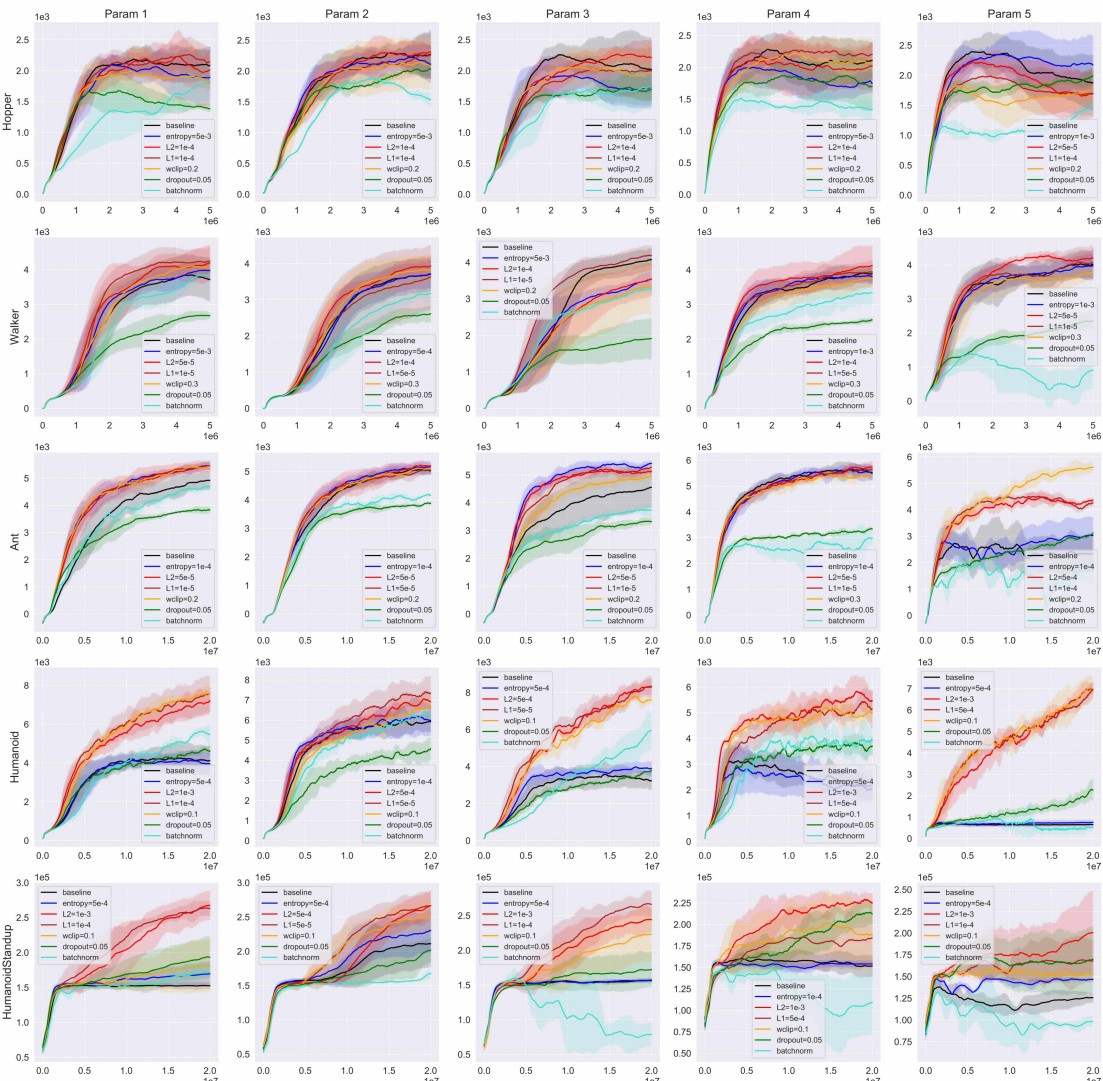

Figure 12: Training curves of PPO regularizations under five randomly sampled hyperparameters.

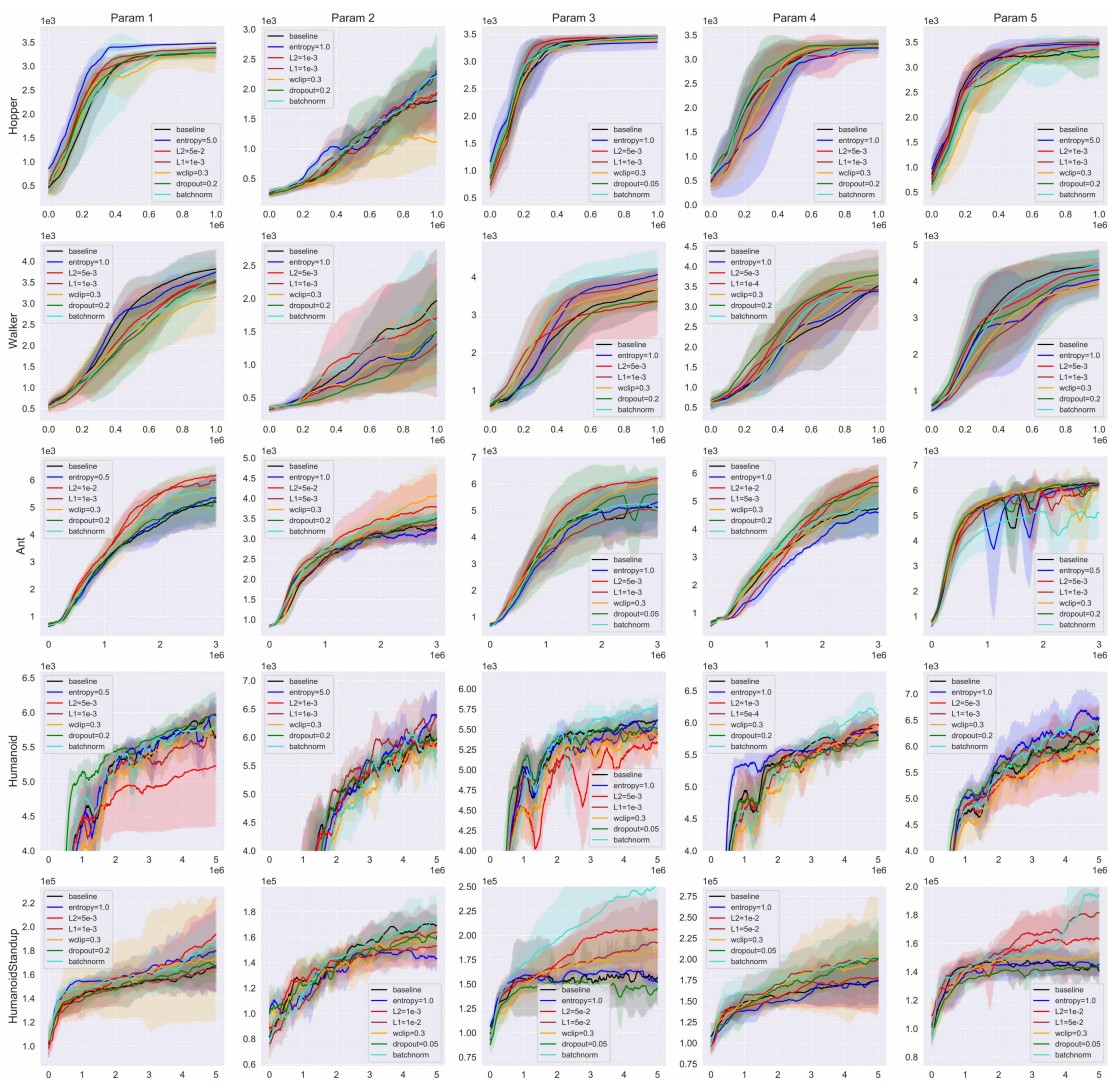

Figure 13: Training curves of SAC regularizations under five randomly sampled hyperparameters.

# L    TRAINING CURVES FOR POLICY VS. VALUE EXPERIMENTS

We plot the training curves with our study in Section 6 on policy and value network regularizations from Figure 14 to Figure 17.

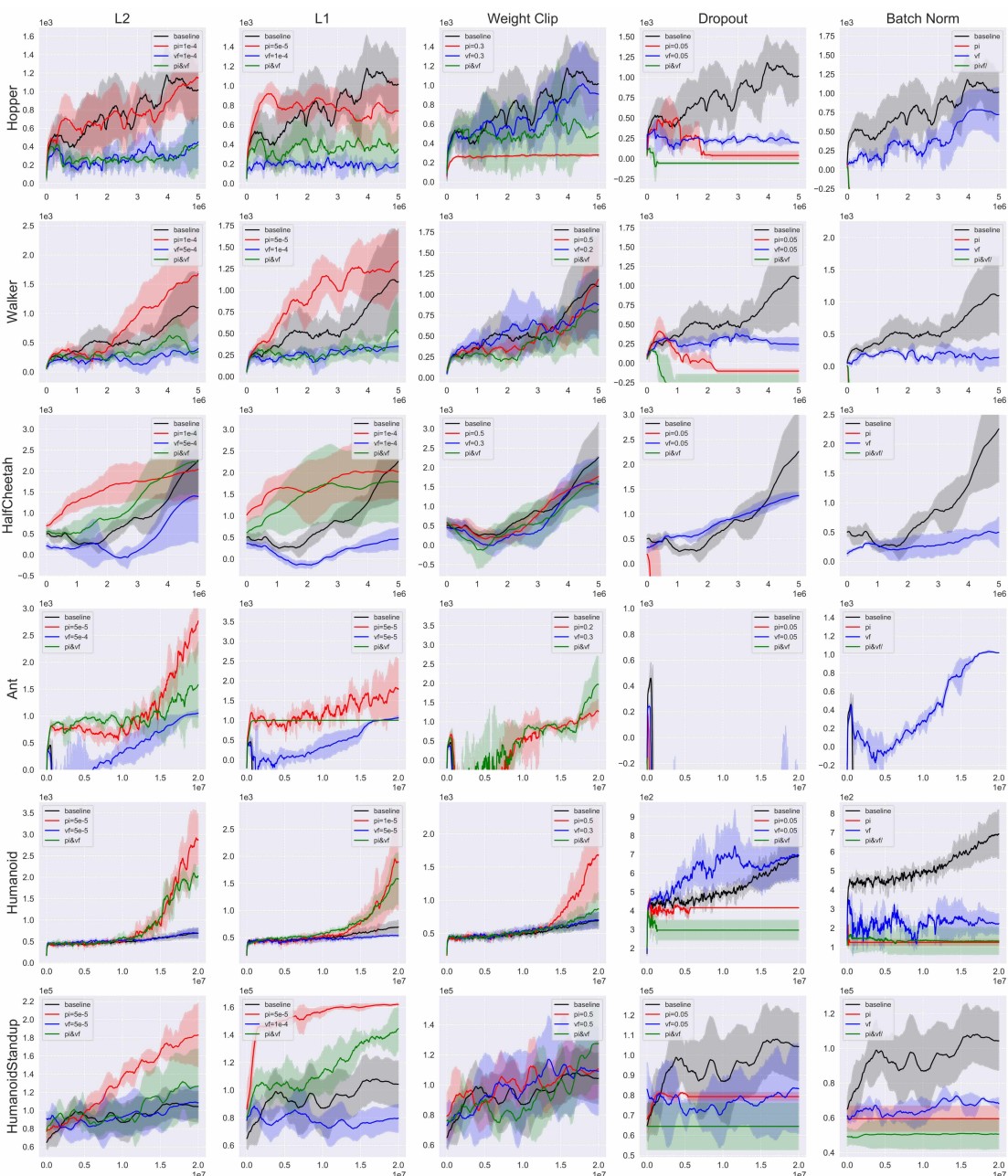

Figure 14: The interaction between policy and value network regularization for A2C. The optimal policy regularization and value regularization strengths are listed in the legends. Results of regularizing both policy and value networks are obtained by combining the optimal policy and value regularization strengths.

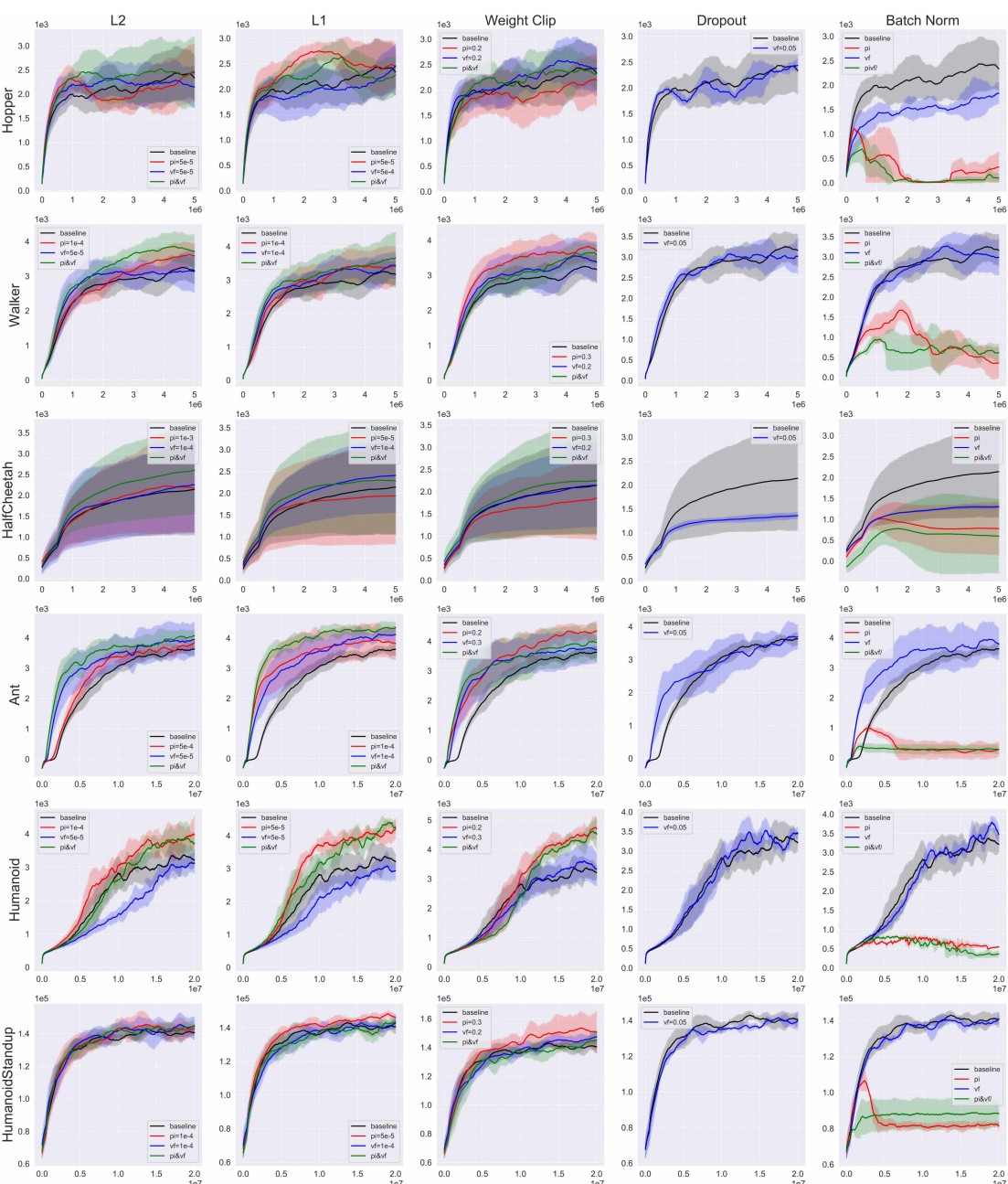

Figure 15: The interaction between policy and value network regularization for TRPO.

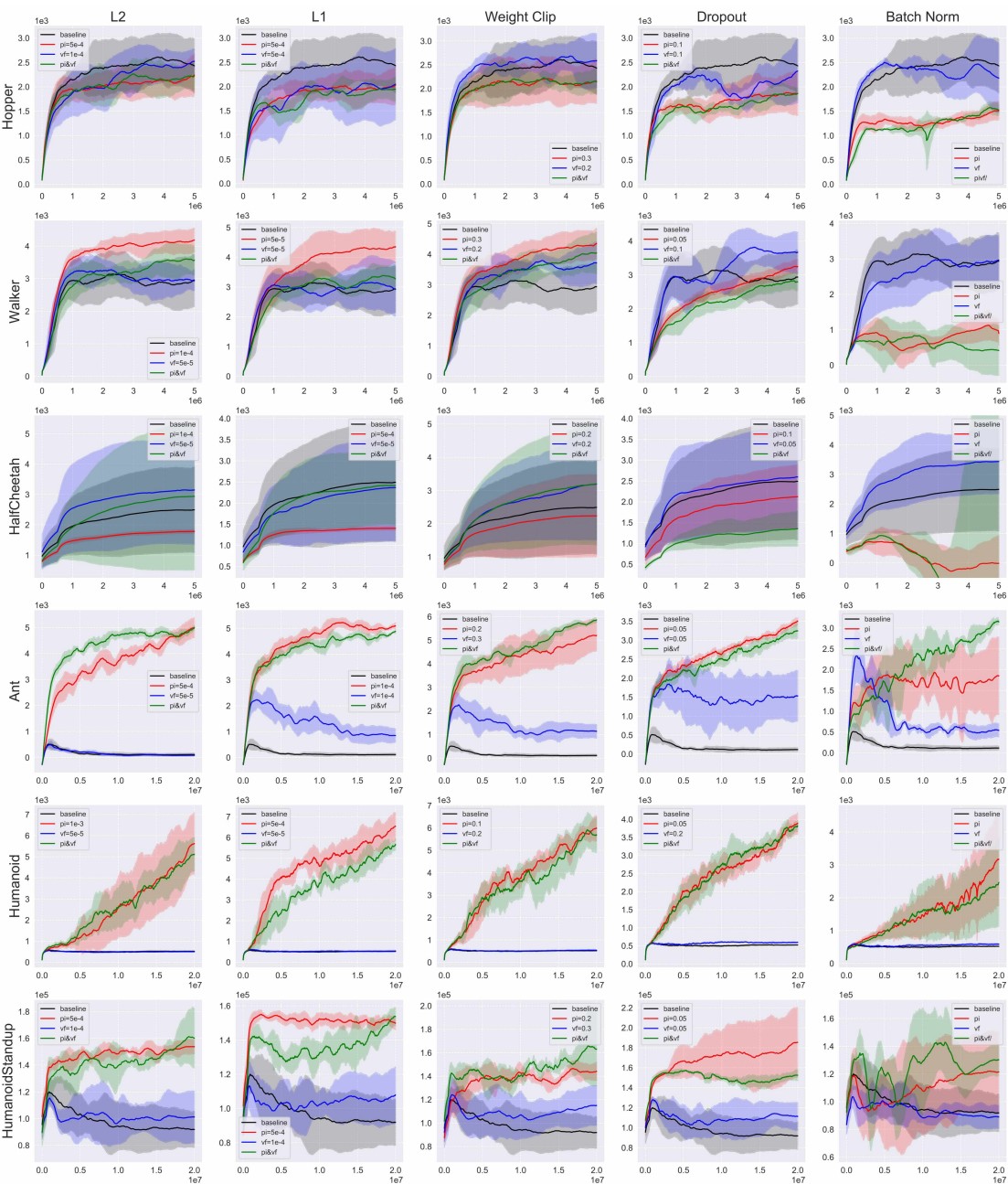

Figure 16: The interaction between policy and value network regularization for PPO.

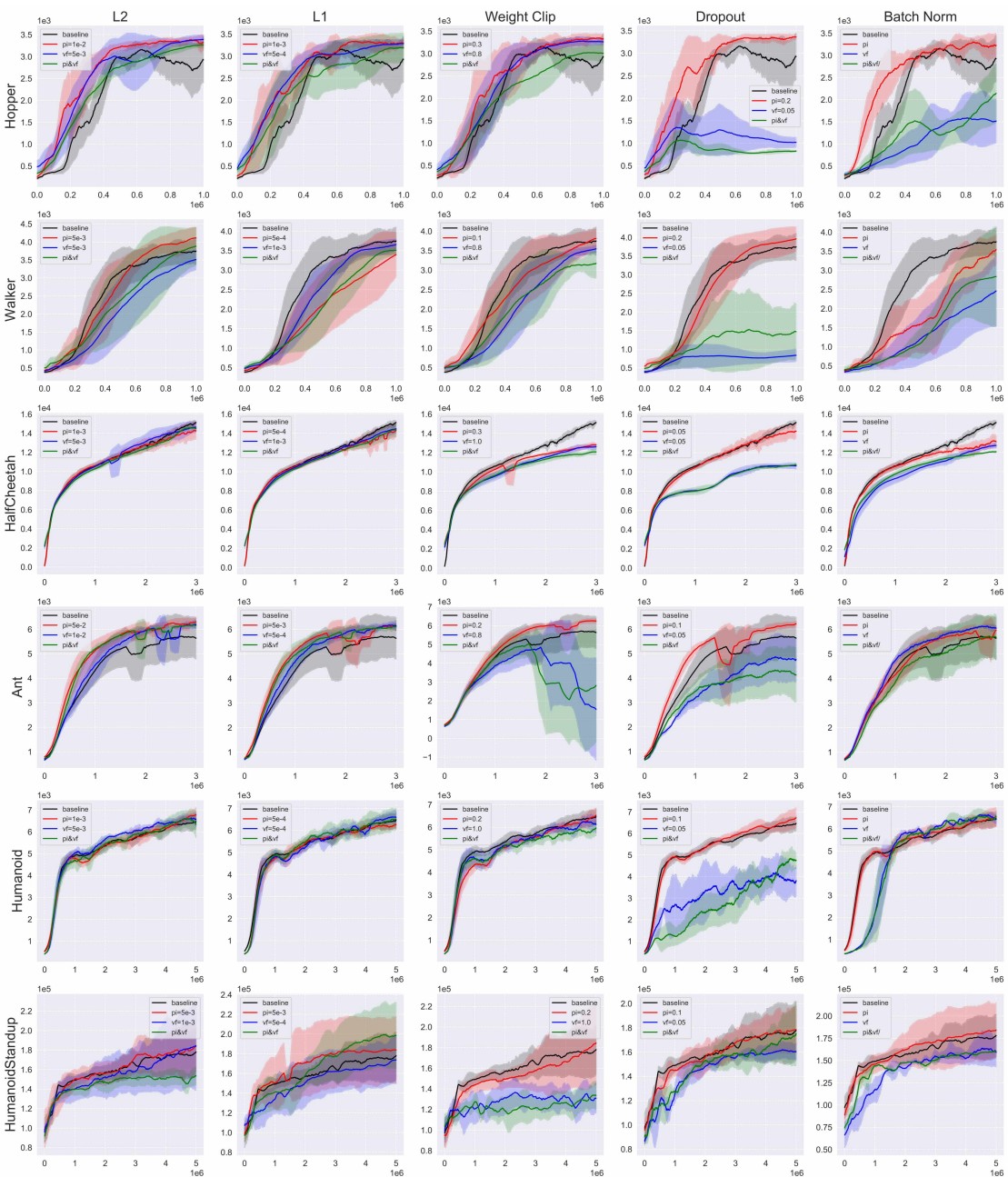

Figure 17: The interaction between policy and value network regularization for SAC.

