# OpenReview forum: "Regularization Matters in Policy Optimization"
_ICLR.cc/2020/Conference — Reject_

### Official Review · AnonReviewer1 · 2019-10-22
**Official Blind Review #1**

**Rating:** 3

**Review:**

An interesting paper on the role of regularization in policy optimization


In this paper, the authors study a set of existing direct policy optimization methods in the field of reinforcement learning. The authors provide a detailed investigation of the effect of regulations on the performance and behavior of agents following these methods.

The authors present that regularization methods mostly help to improve the agents' performance in terms of final scores. Specifically, they show that direct regularizations on model parameters, such as the standard case of L2 or L1 regularization, generally improve the agent performance. They also show that these regularizations, in their study, is more proper than entropy regularization. The authors also show that, in the presence of such regularizations, the learning algorithms become less sensitive to the hyperparameters.

Few comments:
1) The paper is well written and easy to follow. I appreciate it. I found the writing of the paper has a bit of repetition. The authors might find it slightly more proper to remove some of the repetitions (e.g. section 4.2)

2) While I appreciate the clear writing and reasoning in this paper, I might suggest a slight change in the second paraphrase of the intro. I agree with the authors' reason on the first three lines, but I think it would be useful to also emphasize the role of the questions the researchers investigate to answer. I might also add one the main reason that the researchers in the field of DRL have spent less time on regulation or architecture search was their focus on more high-level algorithm design which is in the more immediate step of relevance and specialty to the field of reinforcement learning.

3) I would suggest rephrasing the last two sentences of the second paragraph in related work: "Also, these techniques consider ...". Regularizing the output also regularizes the parameters, I think the authors' point was "directly regularize" the parameters.

4) In the "Entropy Regularization" part of section 3, I guess the Hs has not been defined.

5) Repeated "the" in the last paragraph of section 4.1 (despite it already incorporates the the maximization of)

6) The authors used the term "not converge" multiple times. While it is hard from the plots to see whether the series converges or not, I have a strong feeling that by this term the authors mean the algorithm does not converge to a resealable solution rather than being divergent up to a bandwidth. Maybe clarifying would be helpful.

7) In section 5, the authors study the sensitivity to the hyperparameters. In this section, I had a hard time to understand the role of term 3
"BN and dropout hurts on-policy algorithms but can bring improvement only for the off-policy SAC algorithm." Does it mean that deploying BN, results in a more sensitive algorithm? or it means that the performance degrades (which is a different topic than section 5 is supposed to serve)?

8) In section 7, the authors put out a hypothesis "
However, there is still generalization between samples: the agents are only trained on the limited" but the provided empirical study might not fully be considered to be designed to test this hypothesis. In order to test this hypothesis, the author might be interested in training the models with bigger sample sizes, more training iteration, different function classes, and more fitting in order to test this hypothesis.


9) Section 7 on "Why do BN and dropout work only with off-policy algorithms?" while I agree with the authors on their first reason which is quite commonly known, I might hesitate to make the second statement (2)



Generally, I found this paper an interesting paper and appreciate the authors for their careful empirical study. But I found the contribution of this work to be not significant enough. Most of the statements and claims in this paper are well know in the community, especially among deep learning practitioners. While I acknowledge the scientific value of this study, its concreteness, and appreciate the contribution of this paper, due to the low acceptance rate of this conference, I might be reluctant in accepting this paper.


**Experience Assessment:**

I have published in this field for several years.

**Review Assessment: Checking Correctness Of Derivations And Theory:**

I carefully checked the derivations and theory.

**Review Assessment: Checking Correctness Of Experiments:**

I assessed the sensibility of the experiments.

**Review Assessment: Thoroughness In Paper Reading:**

I read the paper thoroughly.

---

> ### Author Response · Authors · 2019-11-14
> **Response to AnnoReviewer1 [3/3]**
>
>
> References:
> [1] Peter Henderson, Riashat Islam, Philip Bachman, Joelle Pineau, Doina Precup, and David Meger.Deep reinforcement learning that matters. In Thirty-Second AAAI Conference on Artificial Intelli-gence, 2018.
> [2] Islam, R., Henderson, P., Gomrokchi, M., and Precup, D. (2017).  Reproducibility of benchmarked deep reinforcement learning tasks for continuous control.  In ICML 2017 Reproducibility in Machine Learning Workshop.
> [3] https://github.com/jwyang/faster-rcnn.pytorch/blob/master/lib/model/faster_rcnn/resnet.py#L288
> [4] https://github.com/torch/nn/issues/873
> [5] Zafarali Ahmed, Nicolas Le Roux, Mohammad Norouzi, and Dale Schuurmans. Understanding the impact of entropy in policy learning. arXiv:1811.11214, 2018.
> [6] Volodymyr Mnih, Adria Puigdomenech Badia, Mehdi Mirza, Alex Graves, Timothy Lillicrap, TimHarley, David Silver, and Koray Kavukcuoglu. Asynchronous methods for deep reinforcement learning. In International conference on machine learning, pp. 1928–1937, 2016.
> [7] Chenyang Zhao, Olivier Sigaud, Freek Stulp, and Timothy M. Hospedales. Investigating generalisation in continuous deep reinforcement learning. arXiv:1902.07015, 2019.
> [8] Chiyuan Zhang, Oriol Vinyals, Remi Munos, and Samy Bengio.  A study on overfitting in deep reinforcement learning. arXiv:1804.06893, 2018.
> [9] Jesse Farebrother, Marlos C Machado, and Michael Bowling. Generalization and regularization indqn.arXiv preprint arXiv:1810.00123, 2018.
> [10] Karl Cobbe, Oleg Klimov, Chris Hesse, Taehoon Kim, and John Schulman. Quantifying generalization in reinforcement learning. arXiv:1812.02341, 2018.
> [11] https://github.com/openai/baselines
> [12] https://github.com/hill-a/stable-baselines
> [13] https://github.com/ray-project/ray

---

> ### Author Response · Authors · 2019-11-14
> **Response to AnnoReviewer1 [2/3]**
>
>
> (..Continued on A8) For experiments in Figure 7, we vary the number of timesteps/samples seen during training, and present results in the barplot. In the barplot, we find that for regularized models to reach the same level of return as baseline, they only need much fewer samples in training. Note that the return is also on unseen samples. Since the ability to learn from fewer samples is closely related to the notion of generalization, we can conclude that regularized models have better generalization ability than baselines.
>
> Q9. “Section 7 on ‘Why do BN and dropout work only with off-policy algorithms?’ while I agree with the authors on their first reason which is quite commonly known, I might hesitate to make the second statement.”
>
> A9. Batch Normalization layers can be sensitive to input distribution shifts, since the mean and standard deviation statistics depend heavily on the input, and if the input distribution changes too quickly in training, the mapping functions of BN layers can change quickly too, and it can possibly destabilize training. One evidence for this is that in supervised learning, when transferring a ImageNet pretrained model to other vision datasets, sometimes the BN layers are fixed (e.g., see [3,4]) and only other layers are trained. For on-policy algorithms, we always use the samples generated from the latest policy; for off-policy algorithms, the sample distributions are relatively slow-changing since we always draw from the whole replay buffer which holds cumulative data. Like in supervised learning, the faster-changing input distribution for on-policy algorithms could be harmful to BN. We have revised the text to make it more clear in the revision.
>
> We agree that our qualitative analysis is only one of the possible reasons for the BN & on-policy incompatible issue. The real concrete reasons are open to discussion and could be interesting future research. We are happy to remove this part of the analysis if needed.
>
> Q10. Significance of Contribution. “Generally, I found this paper an interesting paper and appreciate the authors for their careful empirical study. But I found the contribution of this work to be not significant enough. Most of the statements and claims in this paper are well know in the community, especially among deep learning practitioners. While I acknowledge the scientific value of this study, its concreteness, and appreciate the contribution of this paper, due to the low acceptance rate of this conference, I might be reluctant in accepting this paper.”
>
> A10. We are glad to see the reviewer finds our work interesting and we thank the reviewer for the acknowledgement on the scientific value of our study. However, we slightly disagree with the statement that most of our statements and findings are already well known in the community. Our reasons are below:
>
> 1. To our best knowledge, our work is the first to study the effects of common regularizers in policy optimization, and no prior publications have experimented with or discussed this issue in detail. Most prior works on RL regularization use or study entropy regularization [5,6], which we have shown to be generally inferior to L2 regularization in our experiments on continuous control tasks; other related works [7,8] study agents’ ability to generalize to new environments, including some on the effects of regularizations [9,10]. Our work is the first to study agent’s performance in the same environment and found common regularizers to be effective, often better than the entropy regularization.
>
> 2. Popular works in this field, such as DQN, TRPO, PPO, SAC, did not consider using common regularizers and did not mention regularization’s effect in their papers. Regularization in RL is an important yet largely ignored issue, as the reviewer also pointed out that most RL works focus on high-level reinforcement learning algorithms.
>
> 3. Popular RL codebases (such as OpenAI Baselines [11], Stable Baselines [12], Ray [13]) do not support our investigated regularization methods other than entropy regularization.
>
> 4. To our best knowledge, our work is the first to discuss whether to regularize policy and/or value network, and the behavior discrepancy between on/off-policy algorithms in terms of BN and dropout, which are very practical problems to consider.
>
> Therefore, we believe our work brings new (and possibly surprising) findings to the community. Despite some practitioners may have some experience in trying common regularizers for RL, our work is the first systematic and comprehensive study that brings the problem and findings to the community, which we believe could guide future research/practices and be a good contribution to ICLR.

---

> ### Author Response · Authors · 2019-11-14
> **Response to AnnoReviewer1 [1/3]**
>
> Thanks for your constructive comments! We try to address your concerns below, and we have uploaded a revision reflecting the changes. For easier reading we pasted some of your comments and please bear with our response length.
>
> Q1. Remove repetition.
>
> A1. We have removed some repetition (of observations) in section 4, for example, “BN and dropout are generally not favorable for the three on-policy algorithms, but they can be useful on SAC (ranking higher than baseline). L1 and weight clipping perform similarly as L2 in TRPO and PPO, better than entropy regularization, but worse in A2C and SAC” in the paragraph of “ranking all regularizations”.
>
> Q2. Suggestion on introduction
>
> A2.
> (1) “I might also add one the main reason that the researchers in the field of DRL have spent less time on regulation...”
> Following your suggestion, we added “Moreover, researchers in deep RL focus more on high-level algorithm designs, which is more closely related to the field of reinforcement learning, and focus less on network training techniques such as regularization” in the second paragraph of intro as a reason why common regularizations were not widely considered.
>
> (2) “It would be useful to also emphasize the role of the questions the researchers investigate to answer.”
> We also added “our results also show that neural network training techniques, such as regularization, can be as important as high-level reinforcement learning algorithms in terms of boosting performance” in the last paragraph of intro (second last sentence), to emphasize our investigated questions’ role in this field.
>
> Q3. “Directly regularize” parameters
> A3. Following your suggestion, we have changed the sentence to “also, these techniques consider regularizing the output of the network, while conventional regularization methods mostly directly regularize the parameters”.
>
> Q4. $H_{s_i}$ not defined.
> We have defined the $H_{s_i}$ in the revision (“$H_{s_i} = -\mathbb{E}_{a_i\sim \pi(a_i|s_i)} \log \pi(a_i|s_i)$, where $(s_i, a_i)$ is the state-action pair”). The left hand side of the equation was mistakenly omitted. Thanks for catching this.
>
> Q5. Repeated “the”.
> We have corrected this typo. Thanks for pointing out.
>
> Q6. Term “Not converge”
> Yes, your guess is correct: by this term we mean “the algorithm does not converge to a reasonable solution”. In the revision, we have changed the term “not converge” to “not converge to a reasonable solution”, and “converge” to “converge to a high level” in corresponding places. Sorry for the confusion.
>
> Q7. “BN and dropout hurts on-policy algorithms but can bring improvement only for the off-policy SAC algorithm.”Does it mean that deploying BN, results in a more sensitive algorithm? or it means that the performance degrades (which is a different topic than section 5 is supposed to serve)?
>
> A7. We mean the performance degrades. This is to confirm the same phenomenon as in section 4 about BN/dropout still holds.
>
> We would like to clarify that the main purpose of section 5 is to confirm our findings in section 4 still hold with multiple hyperparameters, since results in RL are sensitive to hyperparameter changes [1, 2] and thus the conclusions can also be vulnerable to them. In this section, by varying hyperparameter configurations, we found that regularizations can consistently improve the performance with different sampled hyperparameters.
>
> As a side product, this also leads to our additional conclusion: proper regularization can reduce the hyperparameter sensitivity and ease the hyperparameter tuning process of RL algorithms, since they can bring up the performance of baselines with suboptimal hyperparameters to be even higher than baselines with better hyperparameters, as shown in Figure 2 and its corresponding analysis.
>
> Q8. Testing the hypothesis about generalization between samples. “The author might be interested in training the models with bigger sample sizes, more training iteration, different function classes, and more fitting in order to test this hypothesis.”
>
> A8. We have added two sets of experiments to provide evidence for this hypothesis in Appendix G:
>
> For experiments in Figure 5 and 6, we take an already trained model, and then sample multiple trajectories in the environment and evaluate each trajectory’s return. These trajectories are unseen samples during training, since the state space is continuous. The trajectory return’s distributions are plotted in the figures. For baseline, some of the trajectories yield relatively high returns, while others yield low returns, demonstrating the baseline cannot stably generalize to unseen examples; for regularized models, the returns are mostly high and have smaller variance, demonstrating they can more stably generalize to unseen samples.

---

### Official Review · AnonReviewer2 · 2019-10-22
**Official Blind Review #2**

**Rating:** 3

**Review:**

The paper provides an empirical study of regularization in policy optimization methods in multiple continuous control tasks. The paper focuses on the effect of conventional regularization on performance in training environments, not generalization ability to different (but similar) testing environments. Their findings suggest that L2 and entropy regularization can improve the performance, be robust to hyperparameters on the tasks studied in the paper.

Overall, the paper is well written. However, I am leaning to reject this paper because (1) the experimental finding is not well justified (2) the experiments are missing some details and do not provide convincing evidence.

First, the paper does not well justify why regularization methods improve performance in training environments. One potential reason is discussed in Section 7: regularization can improve generalization to unseen samples. However, the improvement can simply due to better hyperparemer optimization. When we introduce more hyperparemers and computation compared to baselines, it’s not surprising to see a better performance, especially in deep RL where using a different seed or using a different implementation can have significant difference in performance [1]. Moreover, it is unclear that inability to generalize to unseen samples is a problem in the continuous control tasks evaluated in the paper. I think the paper should demonstrate that this is indeed a problem. If it is not a problem, why would you expect regularization to help?

There are some missing details which makes it difficult to draw conclusion:
1. How was \sigma_{env,r} computed? Is it the standard error of the mean return, or the standard deviation of the return?
2. What does the average rank mean (in Table 2 and 3)? the average ranking over 5 seeds and all environments? If so, does it make sense to compare these numbers? e.g. Algorithm A with rank 1, 1, 7, 7 and Algorithm B with rank 4, 4, 4, 4 have the same average rank, but totally different performance.
3. The experiment in Figure 3 seems very interesting, however, what’s the conclusion here?
4. Why do you use difference hyperparamer ranges (lambda for L2, L1 and entropy regularization) for different algorithms in appendix A?

Minor comment which does not impact the score:
1. It would have been better if there’s a brief description of each algorithm (before section 4 or in appendix).

[1] Reproducibility of Benchmarked Deep Reinforcement Learning Tasks for Continuous Control


**Experience Assessment:**

I have published one or two papers in this area.

**Review Assessment: Checking Correctness Of Derivations And Theory:**

N/A

**Review Assessment: Checking Correctness Of Experiments:**

I assessed the sensibility of the experiments.

**Review Assessment: Thoroughness In Paper Reading:**

I read the paper at least twice and used my best judgement in assessing the paper.

---

> ### Author Response · Authors · 2019-11-14
> **Response to AnnoReviewer2 [2/2]**
>
> Q3. Missing Details
>
> (1). How was $\sigma_{env,r}$ calculated.
>
> $\sigma_{env, r}$ is the standard deviation of the 5 returns obtained by 5 runs of random seeds. It is calculated as $\sqrt{\frac {\sum_{i=1}^{n} {(r_i - \mu_{env,r})^2}} {n}}$, where $n=5$ and $r_i$ is the return with $i$th seed. In the revision, we have made it clear in the second sentence of the paragraph, that we use standard deviation (not standard error of mean return).
>
> (2). “What does the average rank mean (in Table 2 and 3)? the average ranking over 5 seeds and all environments? If so, does it make sense to compare these numbers? e.g. Algorithm A with rank 1, 1, 7, 7 and Algorithm B with rank 4, 4, 4, 4 have the same average rank, but totally different performance.”
>
> The ranks of mean return ($\mu_{env, r}$), are collected for each environment. Then the average is calculated. In other words, the average ranks are over environments, but not over different random seeds, since we only rank $\mu_{env,r}$s which is already averaged over random seeds. We have made how we calculated average rank more clear in the paragraph “Ranking all regularizations” in section 4 of the revision.
>
> We agree that average rank alone is not the best metric to reflect detailed algorithm behaviors, especially the stability/variance of the algorithm. To better measure the variation, we added three tables (Table 3, 5, and 15) in the paper presenting the standard deviation of ranks. If we look closely, we find L2, L1 and weight clipping actually have relatively smaller stds in most times, whereas baseline, entropy, dropout and BN have larger stds. This means the methods that rank higher also rank (slightly) more stably.
>
> The average rank and rank standard deviations serve as summary statistics of each method. In addition to that, we provided the average percentage of “improving” and “hurting” using our definition. We hope those summarized information could serve as a fair comparison among different regularizers, as fully analyzing the detailed results for each (algorithm, regularizer, environment) tuple would be overwhelming and use too much space. For the detailed behaviors/training curves of each (algorithm, regularizor, environment) tuple, we refer our readers to Figure 1, Appendix C, K and L.
>
> (3). Conclusion of Figure 3.
>
> We had a brief analysis of Figure 3 in the paragraph below it, and we have added more analysis in the revision. There are several observations we can draw: 1) The baseline performance can be either increasing, decreasing or staying roughly the same when the network depth/width increases. 2) Certain regularizations can help with various widths or depths, demonstrating their robustness against these hyperparameter s and ability to ease hyperparameter tuning. 3) Regularizations do not necessarily help more when the network sizes are bigger, contrary to what we might expect: larger networks may suffer more from overfitting and thus regularization can help more. As an example, L2 sometimes helps more with thinner network (TRPO Ant), and sometimes more with wider network (PPO HumanoidStandup).
>
> (4). "Why do you use difference hyperparameter ranges (lambda for L2, L1 and entropy regularization) for different algorithms in appendix A?"
>
> For the three on-policy algorithms (A2C, TRPO, PPO) we use the same tuning range, and the only exception is the off-policy SAC. The reason why SAC’s tuning range is different is that SAC uses a hyperparameter that controls the scaling of the reward signal, while A2C, TRPO, and PPO don’t. In the original implementation of SAC, the reward signals are pre-tuned to be scaled up by a factor ranging from 5 to 100, according to specific environments. Also, unlike A2C, TRPO, and PPO, SAC uses unnormalized reward because if the reward magnitude is small, then, according to the paper, the policy becomes almost uniform. Due to the above reasons, the reward magnitude of SAC is much higher than the magnitude of rewards used by A2C, TRPO, and PPO. Thus, the policy network loss and the value network loss have larger magnitude than those of A2C, TRPO, and PPO, so the appropriate regularization strengths become higher. Considering the SAC’s much larger reward magnitude, in our preliminary experiments, we selected a different range of hyperparameters for SAC before we run the whole experiments.
>
> Q4. Minor comment
> A4. In the revision, we have added brief descriptions of each algorithm in Appendix A.
>
> Thank you again for your review! If you have any further questions we are happy to answer.
>
> [1] Peter Henderson, Riashat Islam, Philip Bachman, Joelle Pineau, Doina Precup, and David Meger.Deep reinforcement learning that matters. In Thirty-Second AAAI Conference on Artificial Intelli-gence, 2018.
> [2] Islam, R., Henderson, P., Gomrokchi, M., and Precup, D. (2017).  Reproducibility of benchmarked deep reinforcement learning tasks for continuous control.  In ICML 2017 Reproducibility in Machine Learning Workshop.

---

> > ### Comment · AnonReviewer2 · 2019-11-14
> > **Quick Reply**
> >
> > I am happy with your response in (1), (3) and (4). For (2), I am still concerned about the average and standard deviation of the ranks. Is there any existing literature using these statistics? It is unclear to me how to interpret the mean and standard deviation of ordinal (or categorical) variables. For example, M1 has an average rank=1.33 and standard deviation=0.47, and M2 has an average rank=1.7 and standard deviation=0.87. We can always compare the numbers (i.e. 1.33 < 1.7), however, can we say M1 is better than M2 (in term of the average rank)? Is M1 statistically significant better than M2? How should we interpret the standard deviation of the ranks?

---

> > > ### Author Response · Authors · 2019-11-15
> > > **About the ranking metric [2/2]**
> > >
> > > p-values:
> > >
> > > Regularization Versus Baseline:
> > > ------------------------------------------------------------------------------------
> > >                          A2C          TRPO          PPO           SAC     TOTAL
> > > ------------------------------------------------------------------------------------
> > > L2                   0.0022      0.0182        0.0000       0.0361    0.0000
> > > L1                   0.0395      0.0104        0.0000       0.1887    0.0000
> > > Weight Clip   0.0947      0.5951        0.0001       0.3923    0.0014
> > > Dropout        0.0001        N/A           1.0000       0.2735    0.0010
> > > BatchNorm   0.0000      0.0000        0.3963       0.0413    0.0077
> > > Entropy         0.0369      0.4499        0.0838       0.4332    0.0070
> > > ------------------------------------------------------------------------------------
> > >
> > > Regularization Versus Entropy:
> > > ------------------------------------------------------------------------------------
> > >                          A2C         TRPO           PPO          SAC         TOTAL
> > > ------------------------------------------------------------------------------------
> > > L2                   0.2515       0.0131         0.0001     0.0677     0.0000
> > > L1                   0.4441       0.0637         0.0026     0.5622     0.0020
> > > Weight Clip   0.8358      1.0000         0.0382     0.8166     0.3225
> > > Dropout         0.0000       N/A            0.1643     0.7650     0.0019
> > > BatchNorm   0.0000      0.0000         0.0326     0.2115     0.0000
> > > ------------------------------------------------------------------------------------
> > >
> > > It can be seen that, in the TOTAL column, all regularization methods’ rankings are statistically significantly different from baseline (p<0.05), and only weight clipping is not significantly different from entropy. For each individual algorithm, the significance is lower, partially due to the fewer number of data points. We can conclude that most of the differences in rankings, when summarized in TOTAL and supported by enough data points, are statistically significant. Note that we did not claim that for each algorithm, every considered regularization is statistically significantly better than entropy/baseline.
> > >
> > > Finally we would like to mention that ranking is only one of our tools for summarizing and  comparing regularizations. From our improvement percentage results and the training curves, we can mostly draw similar observations. If needed, we are happy to move the ranking tables to Appendix, and/or list the complete result tables for each algorithm so the full information is available.
> > >
> > > [1] Katrin Lasinger, René Ranftl, Konrad Schindler, and Vladlen Koltun. Towards Robust Monocular Depth Estimation: Mixing Datasets for Zero-Shot Cross-Dataset Transfer. arXiv:1907.01341, 2019
> > > [2] A. Knapitsch, J. Park, Q.-Y. Zhou, and V. Koltun. Tanks and temples: Benchmarking large-scale scene reconstruction. ACM Transactions on Graphics, 36(4), 2017
> > > [3] https://measuringu.com/mean-ordinal/

---

> > > ### Author Response · Authors · 2019-11-15
> > > **About the ranking metric [1/2]**
> > >
> > > Thank you for your reply!
> > >
> > > There exist some prior works which use average rank, e.g., Table 4 and 5 from [1],  Table 2 from [2]. We believe average rank can be a useful (but certainly not perfect) summary tool when the original performance scores on different datasets/tasks are not comparable. In such cases, taking the mean on the original scores is not meaningful and averaging rankings might be more plausible.
> > >
> > > We found there is quite a lot of interesting debate/analysis of whether it is meaningful to take ordinal data’s mean (e.g., see blogpost at [3]). Standard deviation of rank may be less used and interpretable. We were trying to standard deviations to address the question on variance of algorithms. We’ll be happy to remove the standard deviation tables if needed.
> > >
> > > As for interpretation of your example, in this case maybe we cannot conclude M1 is truly better than M2, but the same problem exists for any other performance score (e.g., error rate in classification) with a standard deviation: if M1 is better than M2 in terms of mean, but they have different stds (either that of M1 or M2 is larger), it is always hard to say which one is truly better. We think the best way to interpret this is to say “M1 is better than M2 in terms of average performance, and/but M1/M2 is more stable in terms of variance”. In our experiment, we found regularizations with higher average rankings (L2, L1, weight clipping) tend to have relatively smaller standard deviations (Table 3 and 4’s caption).
> > >
> > > For statistical significance testing, we use the rank data of regularization methods’ performance on each environment for each algorithm, and we perform t-test for correlated samples (scipy.stats.ttest_rel). Note that we can’t use independent two-sample t-test based only on means and stds, because rank data points between two regularization methods are not independent (e.g., ranks of M1 and M2 on the same environment).
> > >
> > > We calculate p-values based on the ranks on the six hard environments, to tell whether the difference between the ranks of two regularization methods is significant. We use the data from Section 5 because there are more data points than Section 4 for each algorithm. In Section 5, there are 3 hard environments and each environment has 5 hyperparameters, so for each algorithm, there are 3*5=15 ranks for each regularization method (and 4 algorithms * 15 ranks = 60 ranks for the “TOTAL” entry). For example, for SAC, the ranks of L2 are [1 2 1 1 3 3 3 7 2 6 1 6 2 1 2], and the ranks of baseline are [7 6 5 6 2 7 6 2 6 2 5 2 5 7 7], so scipy.stats.ttest_rel gives us the p-value of 0.0361. We compare all regularizations versus the baseline, and we compare conventional regularizations versus entropy regularization, which are the focus of our work. The p-values are shown in the tables below:

---

> ### Author Response · Authors · 2019-11-14
> **Response to AnnoReviewer2 [1/2]**
>
> Thank you for your constructive comments! We have uploaded a revision and we answer your questions below. For easier reading we pasted some of your comments and please bear with our response length.
>
> Q1. “However, the improvement can simply due to better hyperparameter optimization. When we introduce more hyperparameters and computation compared to baselines, it’s not surprising to see a better performance, especially in deep RL where using a different seed or using a different implementation can have significant difference in performance [1].”
>
> A1.
> (1). New Hyperparameter/Hyperparameter Optimization
> We would like to draw the reviewer’s attention to Appendix H. In this section, we restrict the regularization methods to a single strength for each algorithm, across different environments. We show that our main findings in Sections 4 and 5 still hold, and that a shared single strength is already enough for a regularization method to yield better performance in most environments. Regularizations’ effectiveness doesn’t depend on heavy hyperparameter tuning for each environment.
>
> Also, to demonstrate that our results and findings are stable/reproducible, we reran the baseline and L2 regularization in Section 4’s experiments, with the same selected hyperparameters on the 6 MuJoCo environments and all 4 algorithms. We ran the experiments for five new different seeds and obtain $\mu_{env, r}$ (mean of final return over five seeds) and $\sigma_{env,r}$ (standard deviation of final return over five seeds). Out of the 24 (algorithm, environment) pairs, compared with the original experiments, we find that there is only one instance that changes from “improving” to “not improving” (SAC Walker), and another instance changing from “not improving” to “improving” (TRPO Ant). Therefore, the percentage of “improvement” stays the same, and regularization can consistently improve the performance.
>
> (2). New computation
> Negligible computation overhead is induced when a regularizer is applied. Specifically, the increase in training time for BN is ~10%, dropout ~5%, while the other more effective regularizers (L2, L1, Weight clipping, entropy) are all <1%.
>
> (3). Different seeds and implementations
> We would like to emphasize that we tried our best to validate our findings and statements with different configurations, including seeds and training hyperparameters, given the known reproducibility issue [1,2] in RL research: 1) throughout the work, we run each experiment with 5 random seeds, and define “improvement” to be at least at one-std level; 2) in section 5, we vary training hyperparameters to ensure our findings are not specific to the default configuration from the implementation we use; 3) we experiment with single shared hyperparameter in Appendix H; 4) We detailed our hyperparameter search range in Appendix B, open-sourced our code, and checked our results are reproducible in the point #1 above.
>
> Q2. “Moreover, it is unclear that inability to generalize to unseen samples is a problem in the continuous control tasks evaluated in the paper. I think the paper should demonstrate that this is indeed a problem. If it is not a problem, why would you expect regularization to help?”
>
> A2. We have added two sets of experiments to provide evidence for this problem in Appendix G:
>
> First we investigate the agent’s obtained rewards on a set of sampled trajectories. We run PPO Humanoid and TRPO Ant, then summarize the rewards on 100 sampled trajectories and plot the reward distribution in Figure 5 and 6. We find that the trajectories generated by the baselines have very large variance: some of the rewards are high, but others are low. This indicates that the baseline cannot stably and reliably generalize to unseen samples during training.  On the other hand, L2, L1 and weight clipping are able to reduce the variance between trajectories, with most trajectories having high rewards. This suggests that conventional regularization can improve the model’s generalization to larger portion of unseen samples.
>
> Next, we present the results of varying the number of training samples/timesteps in Figure 7. We find that the baseline needs to train on more samples (typically 2x more in Figure 7) to reach the same level of performance as those with certain regularizations. In addition, regularization’s gain over baseline can be larger when the samples are fewer (SAC Ant, TRPO Ant). This demonstrates that the agent’s generalization ability improves with the help of regularization, since it can learn better with relatively fewer samples.

---

> > ### Comment · AnonReviewer2 · 2019-11-14
> > **Quick Reply**
> >
> > Thanks for your reply. I agree with your response in A1.
> >
> > For A2, it is still unclear to me how the paper concludes that the baseline cannot stably generalize to unseen samples based on the experiments in Appendix G.

---

> > > ### Author Response · Authors · 2019-11-14
> > > **Explanation on Appendix G**
> > >
> > > Thanks for your quick reply! We are glad to give more explanations on experiments in Appendix G.
> > >
> > > For experiments in Figure 5 and 6, we take an already trained model, and then sample multiple trajectories in the environment and evaluate each trajectory’s return. These trajectories are unseen samples during training, since the state space is continuous. The trajectory return’s distributions are plotted in the figures. For baseline, some of the trajectories yield relatively high returns, while others yield low returns, demonstrating the baseline cannot stably generalize to unseen examples; for regularized models, the returns are mostly high and have smaller variance, demonstrating they can generalize more stably to unseen samples.
> > >
> > > For experiments in Figure 7, we vary the number of timesteps/samples seen during training, and present results in the barplot. We find that for regularized models to reach the same level of return as baseline, they only need much fewer samples in training. Note that the return is also on unseen samples. Since the ability to learn from fewer samples is closely related to the notion of generalization, we can conclude that regularized models have better generalization ability than baselines.
> > >
> > > We have revised text in the new revision to make it more clear. Finally we would like to add that similar to supervised learning (SL), in RL the model naturally faces the problem of generalization to unseen samples. For RL, the seen samples during training is like the training set in SL, and the sampled trajectories during evaluation is like the test set in SL, and the whole state space of the environment is similar to the input data distribution in SL. The model must learn to generalize from seen examples (training set in SL) to unseen examples (test set in SL) to solve the task, since it cannot traverse the entire state space of the environment (the whole data distribution in SL) during training. Therefore, in RL we might expect regularization to help generalization just as it helps generalization in supervised learning.

---

### Official Review · AnonReviewer3 · 2019-10-23
**Official Blind Review #3**

**Rating:** 6

**Review:**

This paper investigates the use of conventional regularizers for neural networks in the reinforcement learning setting. Contrary to the standard practice of foregoing regularizers in deep RL, the paper finds that their addition can improve the performance of policy gradient algorithms on a standard suite of continuous control tasks. Various regularizers are tried, including l2/l1 regularization, entropy regularization and dropout in a combination with a few standard deep RL algorithms such as TRPO, PPO and SAC. Other experiments also verify the impact of these regularizers on the sensitivity of other hyperparameters and whether regularization should be applied to the value or policy networks.

Overall, I find this paper to be a solid empirical study of regularization in deep reinforcement learning. The experiments are thorough, with various aspects being examined in more detail. I find several of the findings interesting, such as the importance of regularizing solely the policy network and that batch norm/dropout are effective for off-policy methods but not on-policy ones. There were certain points which warranted some further clarification.

I would be willing to increase my score based on the authors' response to the following points:
1) In section 6, the last two sentences ("For A2C, TRPO, and PPO ... so further regularization is unnecessary.") are unclear to me.
	- "rewards are already normalized using running mean filters." I thought that rewards are also normalized for SAC, so I'm not sure how this could explain the difference between the on-policy algorithms and the off-policy ones.
	- "mitigates the overestimation bias...further regularization is unncessary." Could you clarify the connection between regularization and overestimation bias? Related to this point, in section 2 of the paper, it is written that "L2 regularization is applied to the critic Q network because it tends to have overestimation bias (Fujimoto et al., 2018)" but I was not able to find such an explanation in the cited paper though I may have missed it.

2) In section 7, in the paragraph on BN/Dropout, could you clarify the point starting from "1) For both BN and dropout,..."? In particular, which discrepancy between the sampling policy and the optimization policy is being referred to here?

3) Did you consider trying weight decay ("Fixing Weight Decay Regularization in Adam", Loschilov et al. 2018) as a regularizer? Given the success of L2 regularization, it could be possible that weight decay is even more effective.

4) For the hyperparameter sensitivity plots, where one hyperparameter is varied at a time, why are the step sizes for the policy and value networks not included in these experiments? They are usually a critical hyperparameter.


Minor comments and typos:
- On p.5, when defining "hurting", perhaps it could be better to choose a looser definition such as "\mu_r < \mu_b" or "\mu_r - \sigma_r < \mu_b - \sigma_b". This way, there could be a larger distinction between the most effective methods. Currently, both l2 and entropy regularization achieve 0.0% and the next best two regularizers are also under 10%.
- In abstract: "regularizing the policy network is typically enough." Rephrase perhaps? The experiments seem to show that applying a regularizer to only the policy network is better than on both.
- In abstract: "large improvement" -> "large improvements"
- p.2, par. 2: "those regularizations" -> "those regularizers"
- p.3, Weight Clipping: "This greatly stablizes" -> "This greatly stabilizes". This sentence could be rephrased since "This" seems to refer to only weight clipping, but is not the only change in WGANs.
- p.3, Dropout: "regularization technique" -> "regularization techniques"
- p.4, par. 1: "due to more stochasticity" -> "due to increased stochasticity"
- p.4, 2nd to last par.: "during policy update" -> "during policy updates"
- p.5, 2nd to last par.: "sometimes help" -> "sometimes helps", "easier ones baseline is" -> "easier ones the baseline is"
- p.8, 2nd to last par.: "it naturally accepts" -> "they naturally accept", "been shown effective" -> "been shown to be effective"
- p.8, last sentence: "policy network without the value network." -> "policy network but not the value network."




**Experience Assessment:**

I have published one or two papers in this area.

**Review Assessment: Checking Correctness Of Derivations And Theory:**

N/A

**Review Assessment: Checking Correctness Of Experiments:**

I carefully checked the experiments.

**Review Assessment: Thoroughness In Paper Reading:**

I read the paper thoroughly.

---

> ### Author Response · Authors · 2019-11-14
> **Response to AnonReviewer3 [2/2]**
>
> Q3. Weight Decay (Loschilov et al. 2018) and L2 regularization.
>
> A3. Following your suggestion, we implemented “fixed weight decay” (AdamW in the paper) following Loschilov et al. 2018 and compared it with baseline and L2 regularization. We evaluated them with PPO on Humanoid and HumanoidStandup. Similar to L2, we briefly tune the strength of weight decay in AdamW and the optimal one is used. The results are shown in Appendix J (Figure 9).
>
> Interestingly, we found that while both L2 regularization and AdamW can significantly improve the performance over baseline, the performance of AdamW tends to be slightly lower than the performance of L2 regularization.
>
> Q4. Step size changes in hyperparameters sensitivity plots.
>
> A4. We have added experiments on step size (learning rate) variation in Figure 2, Section 5. We find that L2, L1 and weight clipping can consistently improve baseline and make the algorithm less sensitive to learning rate changes. We would like to mention that learning rate is an important hyperparameter we vary in our original experiments in section 5 (Table 4 and 5). (More hyperparameter sampling details in Table 11, Appendix E)
>
>
> Q4. Minor Comments and typos
> Thanks for your detailed comments! We have revised the draft in the revision.
>
> (1). Looser definitions of “hurting”.
> We have added the resulting percentages with definition of hurting being $\mu_r < \mu_b$ in the same paragraph. The results are 11.1% for L2, 16.7% for L1, 22.2% for weight clipping, 55.6% for dropout, 72.2% for BN, and 16.7% for entropy. For reference, if we define hurting $\mu_r - \sigma_r < \mu_b - \sigma_b$, the results are 5.6% for L2, 16.7% for L1, 19.4% for weight clipping, 55.6% for dropout, 69.4% for BN, and 13.9% for entropy. We observe similar trends among different methods with different definitions, and we still observe that regularization rarely hurts, except for BN and dropout (for off-policy algorithms).
>
> (2). Rephrase of “only regularizing policy network”.
> We have replaced “is typically enough” to “is typically the best option” to reflect that it is better than regularizing both policy and value network.
>
> (3)-(4). We have corrected the typos accordingly.
>
> (5). Weight clipping.
> We have corrected the typo, and changed the sentence to “This plays an important role in stabilizing the training of GANs” to indicate it is not the sole factor.
>
> (6)-(11). We have corrected the typos accordingly.
>
> Thanks again for your review! We hope our response addresses your concerns. Any further questions or suggestions are welcome.
>
> [1] https://github.com/haarnoja/sac

---

> ### Author Response · Authors · 2019-11-14
> **Response to AnnoReviewer3 [1/2]**
>
> Thank you for your positive feedback! We address your concerns below and we have uploaded a revision reflecting the changes. For easier reading we pasted some of your comments and please bear with our response length.
>
> Q1a. “‘rewards are already normalized using running mean filters.’ I thought that rewards are also normalized for SAC, so I'm not sure how this could explain the difference between the on-policy algorithms and the off-policy ones.”
>
> A1a. In the official implementation of SAC [1] we are using, the reward is not normalized. Here we were trying to understand why regularizing value networks does not help both on- and off-policy methods, instead of explaining their differences. Our intuition is that observation/reward normalization (for on-policy algorithms) or “clipped double Q learning” (in SAC) can mitigate value overestimation bias, and overestimation bias is related to the need for regularizing critic.
>
> Q1b. “‘mitigates the overestimation bias...further regularization is unnecessary.’ Could you clarify the connection between regularization and overestimation bias? Related to this point, in section 2 of the paper, it is written that "L2 regularization is applied to the critic Q network because it tends to have overestimation bias (Fujimoto et al., 2018)" but I was not able to find such an explanation in the cited paper though I may have missed it.”
>
> A1b. We would like to thank the reviewer for carefully checking the referenced work (Fujimoto et al., 2018). After double check, we found that this work does not explicitly mention the connection between overestimation bias and critic regularization. As a result, we decided to remove explanation in our manuscript, and replace it with empirical analysis on results. We provide our original reasoning for the analysis below:
>
> DDPG (Lillicrap et al., 2015) originally uses L2 regularization in the critic. Fujimoto et al. (2018) found that there is a significant overestimation bias of the Q-network in DDPG. To alleviate this overestimation bias, they proposed TD3 by introducing clipped double Q-learning (SAC inherits this). At the same time, they also removed the critic L2 regularizer. This might imply that if overestimation bias is not a problem, regularization is not needed on critic. However, after double check, we found there is no direct evidence for the relation between overestimation bias and regularizing critic. Thank you for bringing this up and this helped improve our analysis.
>
> Q2. In section 7, in the paragraph on BN/Dropout, could you clarify the point starting from "1) For both BN and dropout,..."? In particular, which discrepancy between the sampling policy and the optimization policy is being referred to here?
>
> A2. The theory behind on-policy policy gradient methods (A2C, TRPO, and PPO) necessitates that the same policy should be used for sampling trajectories (collecting data) and performing policy update (forward/backward of NN), otherwise off-policy issues can harm performance.
>
> For BN and dropout layers, there is a distinction between training and testing mode. For BN, during the test mode, moving average of batch statistics are used as normalization; during the training mode, current batch statistics are used as normalization. For dropout, during the testing mode, all neurons in the neural network are kept; during the training mode, only a random subset of the neurons are kept.
>
> In our experiments, when BN or dropout is applied, the testing mode is used to sample trajectories (collecting data) while the training mode is used for policy update (forward/backward of NN). Therefore, the policy $\pi(a|s)$ parameterized by the network is different between trajectory sampling and policy update, which violates the condition for on-policy algorithms, and causes severe off-policy issues.
>
> This discrepancy still exists even if we use training modes both for sampling trajectories and policy update, because for BN, the batch statistics are different for different batches, and thus the policy network mapping function will be different between sampling and update; for dropout, different subsets of neurons will be dropped out for each iteration, thus the policy will be different between sampling and update. Finally, it is infeasible to train if we both use testing mode for sampling and update. Thus, no matter how we set the training/test mode of BN/dropout, the discrepancy (off-policy issue) still exists.

---

### Decision · Program_Chairs · 2019-12-19

**Decision:**

Reject

**Comment:**

This paper proposes an analysis of regularization for policy optimization. While the multiple effects of regularization are well known in the statistics and optimization community, it is less the case in the RL community. This makes the novelty of the paper difficult to judge as it depends on the familiarity of RL researchers with the two aforementioned communities.

Besides the novelty aspect, which is debatable, reviewers had doubts on the significance of the results, and in particular on the metrics chosen (based on the rank). While defining a "best" algorithm is notoriously difficult, and could be considered outside of the scope of this paper, the fact is that the conclusions reached are still sensitive to that difficulty.

I thus regret to reject this paper as I feel not much more work is necessary to provide a compelling story. I encourage the authors to extend their choice of metrics to be more convincing in their conclusions.

---

> ### Author Response · Authors · 2020-02-27
> **Thank you and new version**
>
> We would like to thank the AC and all reviewers for the constructive feedback! We have since incorporated new metrics (scaled rewards, z-scores), and corresponding statistical significance test in our new version https://arxiv.org/abs/1910.09191v2 . We have also emphasized our novelty aspect and added analytic experiments following the rebuttal.